# Conservation of the glycogen metabolism pathway underlines a pivotal function of storage polysaccharides in *Chlamydiae*

Matthieu Colpaert [1], Derifa Kadouche[1,7], Mathieu Ducatez[1,7], Trestan Pillonel [2], Carole Kebbi-Beghdadi[2], Ugo Cenci[1], Binquan Huang[1,6], Malika Chabi[1], Emmanuel Maes[3], Bernadette Coddeville[1], Loïc Couderc[3], Hélène Touzet[4], Fabrice Bray[5], Catherine Tirtiaux[1], Steven Ball[1], Gilbert Greub [2] & Christophe Colleoni [1✉]

The order *Chlamydiales* includes obligate intracellular pathogens capable of infecting mammals, fishes and amoeba. Unlike other intracellular bacteria for which intracellular adaptation led to the loss of glycogen metabolism pathway, all chlamydial families maintained the nucleotide-sugar dependent glycogen metabolism pathway i.e. the GlgC-pathway with the notable exception of both *Criblamydiaceae* and *Waddliaceae* families. Through detailed genome analysis and biochemical investigations, we have shown that genome rearrangement events have resulted in a defective GlgC-pathway and more importantly we have evidenced a distinct trehalose-dependent GlgE-pathway in both *Criblamydiaceae* and *Waddliaceae* families. Altogether, this study strongly indicates that the glycogen metabolism is retained in all Chlamydiales without exception, highlighting the pivotal function of storage polysaccharides, which has been underestimated to date. We propose that glycogen degradation is a mandatory process for fueling essential metabolic pathways that ensure the survival and virulence of extracellular forms i.e. elementary bodies of Chlamydiales.

[1] University of Lille, CNRS, UMR8576-UGSF-Unité de Glycobiologie Structurale et Fonctionnelle, Lille, France. [2] Institute of Microbiology, University of Lausanne and University Hospital Center, Lausanne, Switzerland. [3] University of Lille, CNRS, Inserm, CHU Lille, Institut Pasteur de Lille, US 41 - UMS 2014 - PLBS, Lille, France. [4] University of Lille, CNRS, Centrale Lille, UMR 9189 - CRIStAL - Centre de Recherche en Informatique Signal et Automatique de Lille, Lille, France. [5] University of Lille, CNRS, USR 3290—MSAP—Miniaturisation pour la Synthèse, l'Analyse et la Protéomique, Lille, France. [6] Present address: State Key Laboratory for Conservation and Utilization of Bio-Resources in Yunnan/School of Agriculture, Yunnan University, Kunming, China. [7] These authors contributed equally: Derifa Kadouche, Mathieu Ducatez. ✉email: christophe.colleoni@univ-lille.fr

*C*hlamydiae forms with Planctomycetes and Verrucomicrobia phyla a very ancient monophyletic group of bacteria known as PVC, which has been recently enriched with additional phyla[1]. The *Chlamydiales* consists of members of the *Chlamydiaceae* family that includes etiological agents of human and animal infectious diseases and at of least eight additional families commonly named "*chlamydia*-related bacteria" or "environmental" chlamydia[2,3].

All Chlamydiales display an obligate intracellular lifestyle due to a massive genome reduction and biphasic development, which includes two major morphological and physiological distinct stages: the elementary body (EB), a non-dividing and infectious form adapted to extracellular survival and the reticulate body (RB), a replicating form located within a membrane surrounded inclusion (for review, see ref. [4]). Following entry into a susceptible cell, the EBs differentiate into RBs within the inclusion. During the intracellular stage, RBs secrete many effector proteins through the type III secretion system and express a wide range of transporters in order to manipulate host metabolism and uptake all the metabolites required for its replication. At the end of the infection cycle, RBs differentiate back into EBs before they are released into the environment[5,6].

Glycogen metabolism loss appears to be a universal feature of the reductive genome evolution experienced by most if not all obligate intracellular bacterial pathogens or symbionts[7,8]. Despite the more advanced genome reduction experienced by the animal-specific *Chlamydiaceae* family (0.9 Mpb) in comparison to other protist-infecting Chlamydiales (2–2.5 Mpb), the glycogen metabolism pathway appears surprisingly preserved[7]. This includes the three-enzymatic activities required for glycogen biosynthesis: GlgC, GlgA, and GlgB[9]. ADP-glucose pyrophosphorylase (GlgC) activity that controls the synthesis and level of nucleotide-sugar, ADP-glucose, dedicated solely to glycogen biosynthesis. Glycogen synthase (GlgA) polymerizes nucleotide-sugar into linear α-1,4-glucan. GlgA activity has a dual function consisting of a primer-independent glucan synthesis and glucan elongation at the non-reducing end of preexisting polymers[10]. When the primer reaches a sufficient degree of polymerization (DP > 15) to fit the catalytic site of the glycogen branching enzyme (GlgB), glycogen branching introduced resulting in the appearance of two non-reducing polymer ends that may be further elongated by GlgA. The repetition of this process results in an exponential increase in the number of non-reducing ends leading to a particle with a 32–40 nm diameter[11].

Until recently, *Waddlia chondrophila* (family *Waddliaceae*) as well as all members of *Criblamydiaceae* could be considered as important exceptions to the universal requirement of *Chlamydiales* for glycogen synthesis. Indeed, genome analysis indicated that ad minima the *glgC* gene was absent from all these bacteria[12-14] and that the function of GlgA was possibly also impaired. Consequently, based on the absence of glycogen reported for all knockout *glgC* mutants in bacteria and plants it was believed that *W. chondrophila* was defective in glycogen synthesis[15,16]. Using transmission electron microscope analysis, we are now reporting numerous glycogen particles within the cytosol of *W. chondrophila* and *Estrella lausannensis* (family *Criblamydiaceae*) EBs, suggesting either another gene encodes a phylogenetically distant protein that overlaps the GlgC activity or an alternative glycogen pathway takes place in these Chlamydiales.

The recent characterization of an alternative glycogen pathway, the so-called GlgE pathway, in *Mycobacterium tuberculosis* and streptomycetes prompted us to probe chlamydial genomes with homolog genes involved in this pathway[17,18]. At variance with the nucleotide-sugar based GlgC pathway, the GlgE pathway consists of the polymerization of α-1,4-glucan chains from maltose-1-P. In Mycobacteria, the latter is produced either from the condensation of glucose-1-P and ADP-glucose catalyzed by a glycosyl transferase called GlgM or from the interconversion of trehalose (α-α-1,1-linked D-glucose) to maltose followed by the phosphorylation of maltose, which is catalyzed by trehalose synthase (TreS) and maltose kinase (Mak) activities, respectively[17]. At the exception of Actinobacteria (i.e., mycobacteria and Streptomycetes), TreS is usually fused to maltokinase (Mak) that phosphorylates maltose into maltose-1-phosphate[19]. Subsequently, maltosyl-1-phosphate transferase (GlgE) mediates the formation of α-1,4-linked polymers by transferring the maltosyl moiety onto the non-reducing end of a growing α-1,4-glucan chain. As in the GlgC pathway, branching enzyme (GlgB) then introduces α-1,6 linkages to give rise to a highly branched α-glucan. The GlgC pathway is found in approximately one-third of the sequenced bacteria and is by far the most widespread and best studied; the GlgE pathway has been identified in 14% of the genomes of α-, β-, γ-proteobacteria while 4% of bacterial genomes possess both GlgC- and GlgE-pathways[19,20].

In order to shed light on metabolism of storage polysaccharide in Chlamydiales, we analyzed 220 genomes (including some genomes assembled from metagenomic data) from 47 different chlamydial species that represent the bulk of currently known chlamydial diversity. A complete GlgE pathway was identified in five chlamydial species distributed in *Criblamydiaceae*, *Waddliaceae*, and *Parachlamydiaceae* families. In this work, we demonstrated that the GlgC pathway is impaired in *Criblamydiaceae* and *Waddliaceae*. The complete biochemical characterization of the GlgE pathway in *Estrella lausannensis* (family *Criblamydiaceae*) and *Waddlia chondrophila* (family *Waddliaceae*) isolated, respectively, from water in Spain[21,22] and from the tissue of an abortive bovine fetus[23,24] is reported. Thus, despite the intensive reductive genome evolution experienced by these intracellular bacteria our work shows that glycogen biosynthesis is maintained in all Chlamydiales and suggests a hitherto understudied function of storage polysaccharides and oligosaccharides in the developmental cycle of all *Chlamydiales*.

## Results

**Two different glycogen metabolic pathways are identified in the *Chlamydiae* phylum**. To gain insight into Chlamydiae's glycogen metabolism, we analyzed 220 genomes from 47 different chlamydial species. As illustrated in Fig. 1a, the synthesis of linear chains of the synthesis of linearboth ADP-glucose pyrophosphorylase (GlgC) and glycogen synthase (GlgA) activities in the GlgC pathway while the GlgE pathway relies on trehalose synthase (TreS), maltokinase (Mak), and maltosyl-1-phosphate transferase (GlgE). The formation of α-1,6 linkages (i.e., branching points) and glycogen degradation are catalyzed by a set of similar enzymes in both pathways that include glycogen branching enzyme isoforms (GlgB/GlgB2) and glycogen phosphorylases isoforms (GlgP/GlgP2), glycogen debranching enzymes (GlgX) and glycogen phosphorylases isoforms The genomic database used in this study (https://chlamdb.ch) includes genomes from both cultured and uncultured Chlamydiae species that cover the diversity of the chlamydiae phylum (Fig. 1b). It should be stressed out that several families and genus level-lineages encompass exclusively uncultured Chlamydiae species. As a consequence, derived genomes from metagenomic data have been carefully re-annotated and subjected to various quality criteria, such as the proportion of core genes as previously reported[3]. Comparative genomics clearly underlined the high prevalence of a complete GlgC pathway in most *Chlamydiales*, including all members of the *Chlamydiaceae* family, which has undergone massive genome reduction (identified by the letter "d" on Fig. 1b)

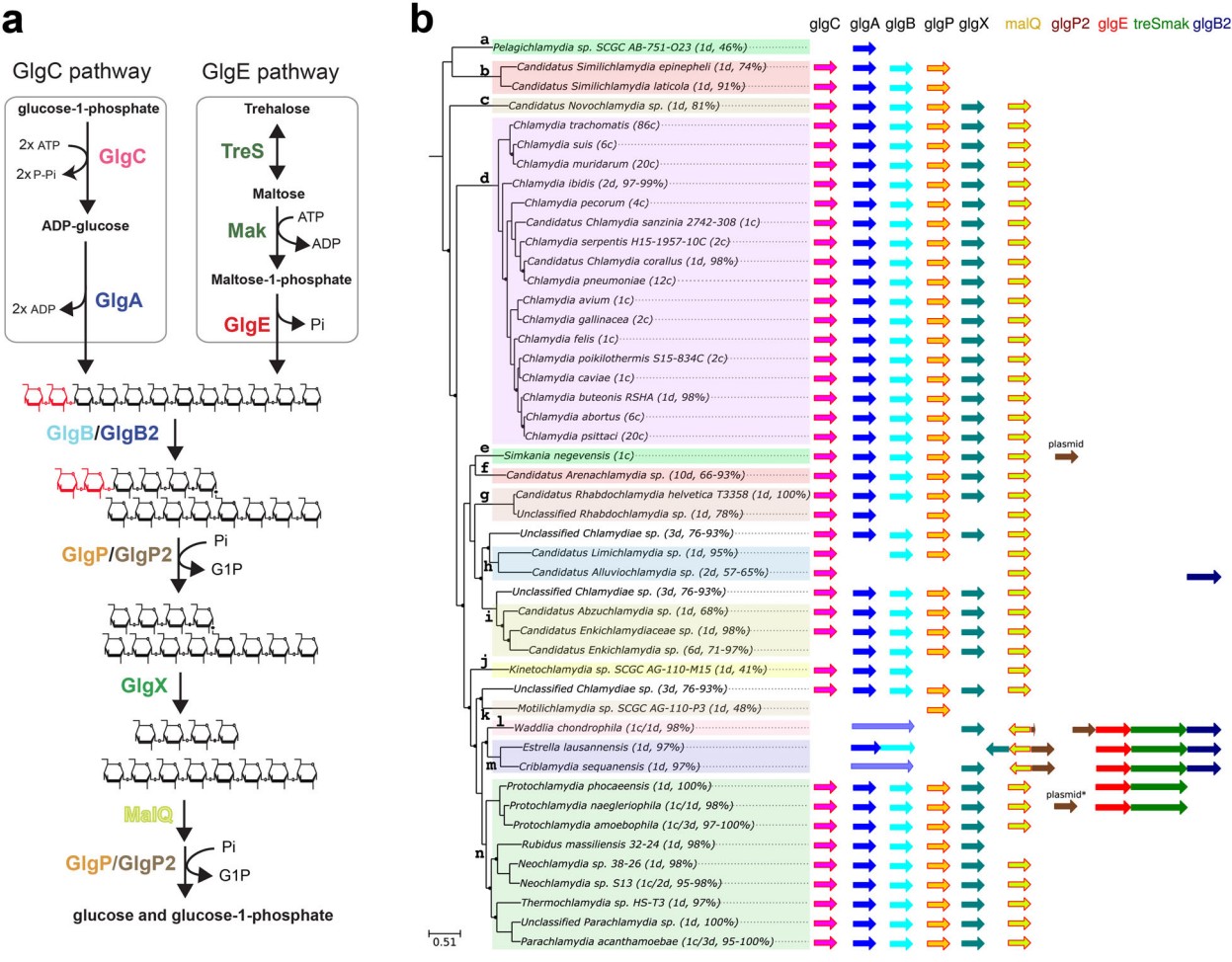

**Fig. 1 Comparative genomic analysis of glycogen-metabolizing genes among *Chlamydiae* phylum. a** GlgC- and GlgE-paths represent the main routes of glycogen biosynthesis in prokaryotes. The formation of linear chains of glucosyl units joined by α-1,4 linkages depends on the coupled actions of ADP-glucose pyrophosphorylase (GlgC)/glycogen synthase (GlgA) activities in GlgC-path whereas it relies on the combined actions of trehalose synthase (TreS)/maltokinase (Mak)/maltosyltransferase (GlgE) in GlgE-path. The iteration of glucan synthesis and branching reactions catalyzed by branching enzyme isoforms (GlgB and glgB2) generate a branched polysaccharide. Both α-1,4 and α-1,6 glucosidic linkages are catabolized through synergic actions of glycogen phosphorylase isoforms (GlgP and GlgP2), debranching enzyme (GlgX), and a-1,4-glucanotransferase (MalQ) into glucose-1-phosphate and glucose. **b** Phylogenic tree of cultured and uncultured Chlamydiae. For each species of the families: a, Ca. *Pelagichlamydiaceae*; b, Ca. *Paralichlamydiaceae*; c, Ca. *Novochlamydiaceae*; d, *Chlamydiaceae*; e, *Simkaniaceae*; f, Ca. *Arenachlamydiaceae*; g, *Rhabdochlamydiaceae*; h, Ca. *Limichlamydiaceae*; i, Ca. *Enkichlamydiaceae*; j, Ca. *Kinetochlamydiaceae*; k, Ca. *Motilichlamydiaceae*; l, *Waddliaceae*; m, *Criblamydiaceae*; n, *Parachlamydiaceae*, the number of draft (d) or complete (c) genomes and genome completeness expressed in a percentage are indicated between brackets. Homologous genes of the GlgC- and GlgE-pathways are symbolized with colored arrows. The glgP2 gene was identified on the plasmid of *S. negevensis* and is also present in one of the two available *P. neagleriophila* genomes.

as well as in in the most deeply branching families such as candidatus *Pelagichlamycidiaceae* ("a") and candidatus *Parili-chlamydiaceae* ("b"). We noticed that the *glg* genes are at least 10 kbp apart with a notable exception for *glgP* and *glgC*, which are separated by one or two genes in most cases. Caution must be taken in interpreting the gaps in glycogen metabolism pathways of several uncultivated chlamydiae, which likely reflect the fact that many of those genomes are incomplete genomes derived from metagenomic studies (see percentages in brackets in Fig. 1b). Considering that the GlgC pathway is highly conserved in nearly all sequenced genomes of the phylum, missing genes probably reflect missing data rather than gene losses. It is interesting to note that there is some uncertainty about the presence of the *glgC* gene in candidatus *Enkichlamydia* genome ("j"), as a complete set of glycogen-metabolizing enzymes were recovered except for the gene encoding for ADP-glucose pyrophosphorylase (glgC). This gene is missing from six independent draft genomes

estimated to be 71–97% complete, suggesting either the loss of *glgC* gene or that *glgC* gene is located in a particular genomic region (e.g., next to repeated sequences) that systematically led to its absence from genome assemblies. Another unexpected result concerns both *Waddliaceae* ("l") and *Criblamydiaceae* ("m") families that encompass *Waddlia chondrophila*, *Estrella lausannensis*, and *Criblamydia sequanensis*. Genomic rearrangements caused a sequence of events leading to (i) the deletion of both *glgC* and *glgP* genes, (ii) the fusion of *glgA* with the *glgB* gene, (iii) the insertion of the *glgP2* gene encoding a glycogen phosphorylase isoform at the vicinity of *malQ* gene. It should be noted that an homolog of *glgP2* has also been identified on the plasmids of *S. nevegensis* and *P. naegleriophila*. In *W. chondrophila*, another insertion of *glgP2* occurred downstream to the GlgE operon, which may be correlated with partial deletion of *glgP2* at the vicinity of *malQ* (Fig. 1b). The parsimonious interpretation of *glgC* and *glgP* deletion and *glgAglgB* fusion is that a

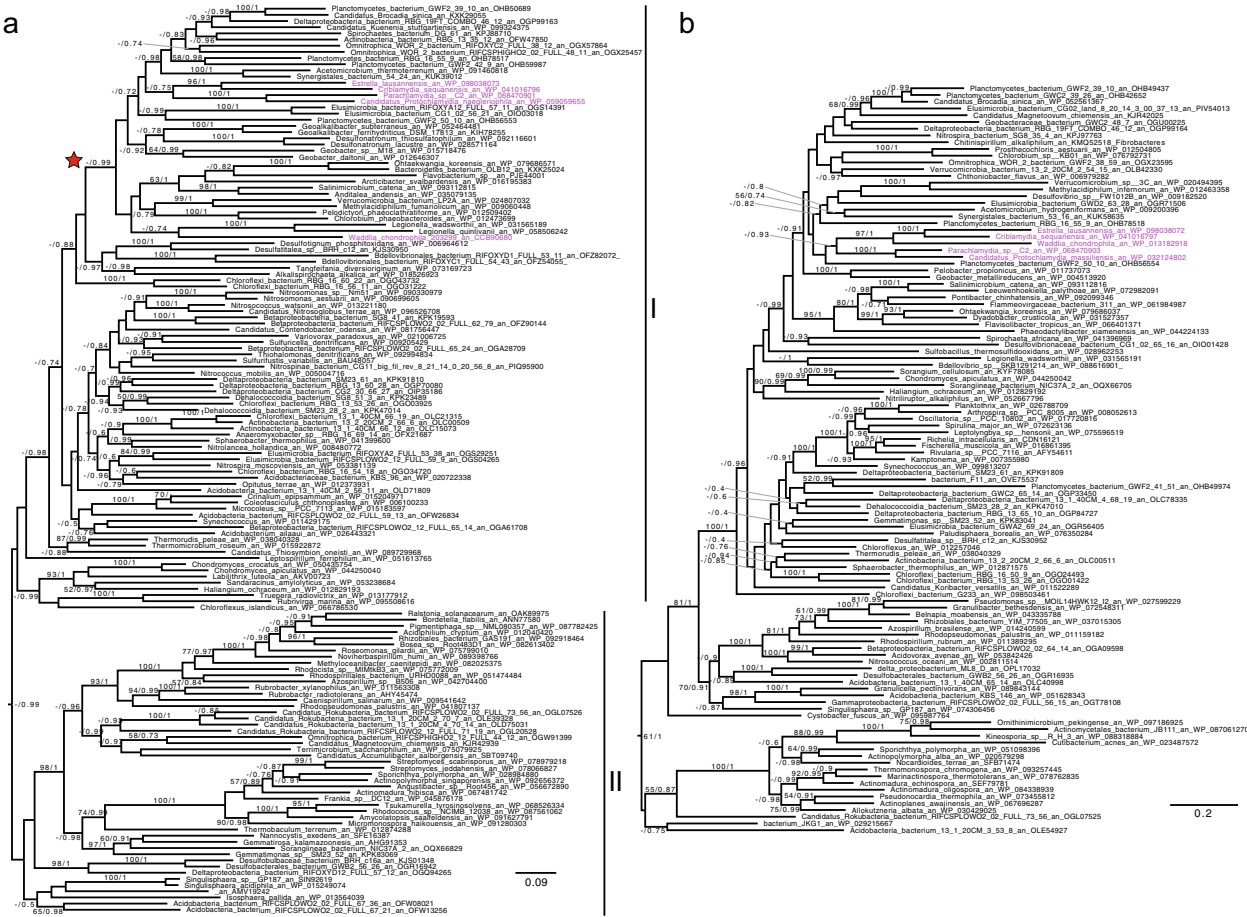

**Fig. 2 Phylogenetic analysis of GlgE and TreS-Mak.** Both phylogenetic trees of GlgE (**a**) and TreS-Mak (**b**) were performed with Phylobayes under the C20 + Poisson model. We then mapped onto the nodes ML bootstrap values obtained from 100 bootstrap repetitions with LG4X model (left) and Bayesian posterior probabilities (right). Bootstrap values >50% are shown, while only posterior probabilities >0.6 are shown. The trees are midpoint rooted. The *Chlamydiales* are displayed in purple. The scale bar shows the inferred number of amino-acid substitutions per site.

single-deletion event led to the loss of DNA fragment bearing *glgP* and *glgC* genes between *glgA* and *glgB*. However, despite many variations, the genomic configuration compatible with this parsimonious hypothesis was never observed in extant *Chlamydiae* (Supplementary Table 1). Rather, such genomic rearrangements are associated with a novel glycogen pathway based on the GlgE operon described in mycobacteria and also observed in *Prototochlamydia naegleriophila* and *Protochlamydia phocaeensis* (syn. *Parachlamydia* C2). All three genes are clustered in the classical unfused *glgE-treSmak-glgB2* operon arrangement in *Waddliaceae* and *Criblamydiaceae*, while the *glgB2* gene is missing in the *Parachlamydiaceae* operons (Fig. 1b). The occurrence of the GlgE pathway restricted to *Parachlamydiaceae*, *Waddliaceae*, and *Criblamydiaceae* families beg the question of its origin in Chlamydiales. To get some insight on this issue, phylogenetic trees of TreS-Mak and GlgE have been inferred using the phylobayes method (Fig. 2). The GlgE phylogeny shows that even if the *Chlamydiae* sequences are split into two with *W. chondrophila* on one side and the other sequences on the other side, which reflects likely lateral gene transfer events with other bacteria, chlamydial *glgE* sequences might still be monophyletic since the only strongly supported node (marked as red star) with a posterior probability (pp) higher than 0.95 (pp = 0.99) unifies all chlamydiae sequences (Fig. 2a), which has also been confirmed using the LG model (Supplementary Data 1). The phylogeny analysis highlights that GlgE sequences can be classified into classes I and II, comprising *Chlamydiales* and *Actinomycetales* (i.e., mycobacteria,

Streptomycetes), respectively. For Tres-Mak phylogeny (Fig. 2b), chlamydial Tres-Mak sequences cluster together, suggesting a common origin, however, with a low statistical support (pp = 0.93). Although the origin of GlgE operon cannot be pinpointed in our phylogenetic analysis, conceivable scenarios are that either (i) the GlgE operon reflects vestigial metabolic function of the ancestral chlamydiae and then has been lost in most families or (ii) this operon was acquired by a lateral gene transfer event from a member of the PVC phylum by the common ancestor of *Parachlamydiaceae*, *Waddliaceae*, and *Criblamydiaceae* families.

**The classical GlgC pathway is not functional in *E. lausannensis* and *W. chondrophila*.** To further investigate whether his-tagged recombinant proteins GlgA–GlgB of *E. lausannensis* and *W. chondrophila* are functional, glycogen synthase activities at the N-terminus domain were assayed by measuring the incorporation of labeled $^{14}C$-glucosyl moiety from ADP- or UDP-$^{14}C$-glucose onto glycogen and by performing a specific nondenaturing PAGE or zymogram to visualize glycogen synthase activities. After separation on native-PAGE containing glycogen, recombinant proteins were incubated in the presence of 1.2 mM ADP-glucose or UDP-glucose, glycogen synthase activities are visualized as dark activity bands after soaking gels in iodine solution (Fig. 3).

Enzymatic assays and zymogram analyses show that the glycogen synthase domain of the chimeric GlgA–GlgB of *W. chondrophila* (hereafter GlgA–GlgB-WC) is functional but highly

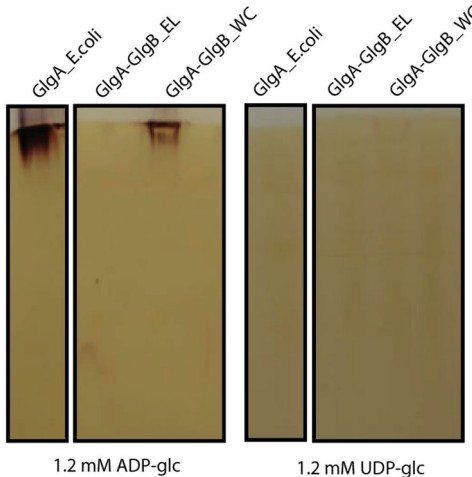

**Fig. 3 Zymogram analysis of glycogen synthase activities.** Total crude extracts of the recombinant proteins of GlgA of *Escherichia coli* (GlgA_E. coli), GlgA–GlgB of *E. lausannensis* (GlgA-GlgB_EL), and *W. chondrophila* (GlgA-GlgB_WC) were separated by native-PAGE containing 0.6% (w/v) glycogen. The native gels were then incubated with 1.2 mM ADP-glc or 1.2 mM UDP-glc. Glycogen synthase activities are seen after iodine staining as dark bands.

specific for ADP-glucose (0.70 nmol of incorporated glucose. $min^{-1}.mg^{-1}$) and has little to no activity using UDP-glucose as substrate. As predicted, the activity of the truncated glycogen synthase in *E. lausannensis* was not detected on activity gels or during enzymatic assays (Supplementary Fig. 1a).

We further investigated whether the branching activity domain at the carboxyl terminal of chimeric protein GlgA–GlgB of *W. chondrophila* (GlgA–GlgB-WC) was functional. To check this, the same chimeric GlgA–GlgB-WC sample previously analyzed was incubated with ADP-glucose (3 mM) and maltoheptaose (10 mg. $mL^{-1}$) overnight. Subsequently, the appearance of branching point (i.e., α-1,6 linkages) onto growing linear glucans can be specifically observed by the resonance of protons onto carbon 6 at 4.9 ppm using proton-NMR analysis. However, as depicted in Supplementary Fig. 1c, we did not observe any signal, suggesting that branching enzyme activity domain is not functional despite an active glycogen synthase domain. This result is consistent with several reports indicating that the amino-acid length at the N-terminus of branching enzyme affects its catalytic properties[25–27]. In regard to this information, the glycogen synthase domain extension located at the N-terminus prevents, probably, the branching enzyme activity of GlgA–GlgB. Thus α-1,6 linkages or branching points are likely to be the result of the GlgB2 isoform activity found in both instances. Altogether, these data strongly suggest that the classical GlgC pathway is not functional in both *Waddliaceae* and *Criblamydiaceae* families.

**GlgE-like genes of *E. lausannensis* and *W. chondrophila* encode α-maltose-1-phosphate: 1,4-α-D-glucan 4-α-D-maltosyltransferase.** Based on phylogenetic analysis of GlgE, both GlgE of mycobacteria (*Actinobacteria*) and *Chlamydiales* are phylogenetically distant (Fig. 2a). GlgE of *M. tuberculosis* displays 43–40% of identity with GlgE-like sequences of *E. lausannensis* and *W. chondrophila*, respectively. Because GlgE activity belongs to the large and diversified Glycosyl Hydrolase 13 family consisting of carbohydrate active enzymes with quite diverse activities such as α-amylases, branching enzymes, debranching enzymes[28], we undertook to demonstrate that these enzymes displayed catalytic properties similar to those previously described for GlgE of mycobacteria. Histidine-

tagged recombinant proteins of GlgE of *Estrella lausannensis* (hereafter GlgE-EL) and *Waddlia chondrophila* (hereafter GlgE-WC) were expressed and further characterized (Supplementary Fig. 2). As described in previous studies, GlgE of *Mycobacteria* mediates the reversible reaction consisting of the release of maltose-1-phosphate in the presence of orthophosphate and α-glucan polysaccharide. Both GlgE-EL and GlgE-WC were incubated in presence of glycogen from rabbit liver and orthophosphate. After overnight incubation, reaction products were analyzed on thin-layer chromatography and sprayed with oricinol-sulfuric acid (Fig. 4a). A fast migration product capable of interacting with orcinol-sulfuric acid was clearly synthesized in crude extract (CE), in washing # 3 (W3), and in the purified enzyme fraction (E1) of the GlgE-EL sample. A barely visible product is only observed in the purified fraction (E1) of GlgE-WC. The compound produced by GlgE-EL in presence of glycogen and orthophosphate was further purified through different chromatography steps and subjected to mass spectrometry and proton-NMR analyses (Fig. 4b, c). The combination of these approaches confirms that GlgE of *E. lausannensis* as well as *W. chondrophila* (Supplementary Fig. 3) catalyzes the formation of a compound with a molecular weight of 422 Da (Fig. 4c) corresponding to α-maltose-1-phosphate, as shown on the proton and phosphorus spectra (Fig. 4b). In order to carry out enzymatic characterization of GlgE activities, identical purification processes were scaled up to purify enough M1P, free of inorganic phosphate and glucan.

**Kinetic parameters of GlgE activity of *E. lausannensis* in the biosynthetic direction.** Because the his-tagged recombinant GlgE-WC expresses very poorly and the specific activity of GlgE-WC was ten times lower than GlgE-EL, kinetic parameters were determined in the synthesis direction, i.e., the transfer (amount) of maltosyl moieties onto non-reducing ends of glucan chains, exclusively for GlgE-EL. Transfer reactions are associated with the release of inorganic phosphate that can be easily monitored through the sensitive malachite green assay. Thus, under variable M1P concentrations and using fixed concentrations of glycogen or maltoheptaose, the GlgE-EL activity displays allosteric behavior indicating positive cooperativity, which has been corroborated with Hill coefficients that were above 1 (Fig. 5a, b). In agreement with this, the molecular weight of native GlgE-EL determined either by size exclusion chromatography or by native-PAGE containing different acrylamide concentrations (5, 7.5, 10, and 12.5%) indicates an apparent molecular weight of 140–180 kD, respectively, corresponding to the formation of dimer species while no monomer species of 75 kD were observed (Fig. 5e, f). The enzyme exhibited $S_{0.5}$ values for M1P that vary from $0.16 \pm 0.01$ mM to $0.33 \pm 0.02$ mM if DP7 and glycogen are glucan acceptors, respectively. However, using M1P at saturating concentration, GlgE-EL displays Michaelis kinetics ($n_H$ close to 1) indicating a non-cooperative reaction (Fig. 5c, d). In such experimental conditions, the apparent $K_m$ values for glycogen and DP7, $2.5 \pm 0.2$ mg.$mL^{-1}$ and $3.1 \pm 0.2$ mM, respectively, were similar to the apparent $K_m$ value of glycogen synthase (GlgA) that synthesizes α-1,4 linkages from ADP-glucose[29].

**De novo glycogen synthesis: GlgE activity enables the initiation and elongation of glucan.** At variance with eukaryotic glycogen synthase, prokaryotic glycogen synthase (GlgA) does not require the presence of a short α-1,4-glucan or primer to initiate glycogen biosynthesis[10]. In absence of GlgA and GlgC activity in *E. lausannensis* and in the absence of GlgC and thus of ADP-glucose supply in *W. chondrophila*, this raised the question of the ability

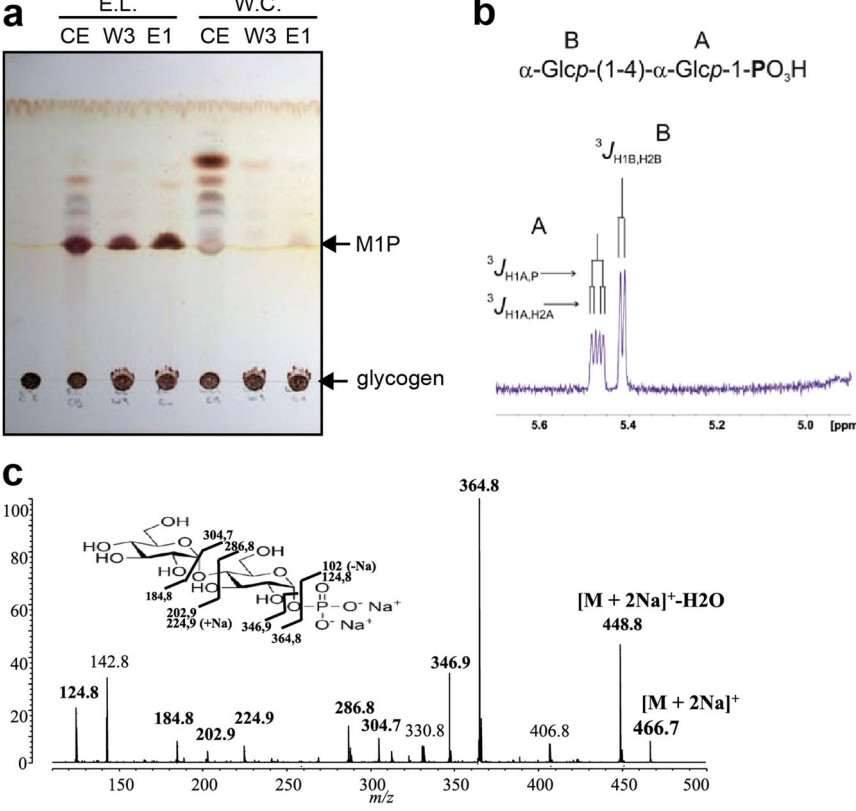

**Fig. 4 Characterization of compounds released by recombinant GlgE of *Estrella lausannensis*. a** Both histidine-tagged recombinant GlgE-EL and GlgE-WC proteins were purified and incubated in presence of glycogen and inorganic phosphate. The overnight reaction products from crude extract (CE), third washing step (W3) purified enzymes (E1) were subjected to thin-layer chromatography analysis. Orcinol-sulfuric spray reveals a notable production of M1P with recombinant GlgE-EL, which is less visible with recombinant GlgE-WC. **b** Part of 1D-$^1$H-NMR spectrum of maltoside-1-phosphate. α-anomer configuration of both glucosyl residues was characterized by their typical homonuclear vicinal coupling constants ($^3J_{H1A,H2A}$ and $^3J_{H1B,H2B}$) with values of 3.5 and 3.8 Hz, respectively. A supplementary coupling constant was observed for α-anomeric proton of residue A as shown the presence of the characteristic doublet at 5.47 ppm. This supplementary coupling constant is due to the heteronuclear vicinal correlation ($^3J_{H1A,P}$) between anomeric proton of residue A and phosphorus atom of a phosphate group, indicating that phosphate group was undoubtedly O-linked on the first carbon of the terminal reducing glucosyl unit A. The value of this $^3J_{H1A,P}$ was measured to 7.1 Hz (Table 1). **c** MS-MS sequencing profile of M1P. The molecular ion [M + 2Na]$^+$ at m/z 466.7 corresponding to M1P + 2 sodium was fractionated in different ions. Peak assignments were determined according to panel incrusted in (**c**).

---

**Table 1 $^1$H chemical shifts (ppm) of anomeric protons and their first vicinal coupling constants (J Hz) recorded at 300 K in D$_2$O displayed on Fig. 4b.**

| Residue | H1 | $^3J_{H1,H2}$ | $^3J_{H1,P}$ |
|---|---|---|---|
| B α-Glcp(1->4) | 5.403 | 3.8 | – |
| A α-Glcp(1->P) | 5.459 | 3.5 | 7.1 |

of GlgE activities to substitute for GlgA with respect to the priming of glycogen biosynthesis. To establish whether GlgE activities are able to prime glucan synthesis, both his-tagged GlgE-EL (3.51 nmol of Pi released.min$^{-1}$) and GlgE-WC (1.38 nmol of Pi released.min$^{-1}$) were incubated with 1.6 mM M1P in the presence of 5 mM of various glucan chains with a degree of polymerization (DP) of 1 to 7. Identical incubation experiments were conducted with GlgE recombinant proteins except M1P was omitted in order to appreciate α-1,4-glucanotransferase or disproportionnating activity (Fig. 6 and Supplementary Figs. 4 and 5).

After incubation, the reduced-ends of glucan chains were labeled with fluorescent charged probe (APTS) and separated according to their degree of polymerization by capillary electrophoresis. We noticed that the C1 phosphate group prevented the labeling of M1P with fluorescent probe. Nevertheless, the level of maltose released from M1P due to the spontaneous dephosphorylation during the experiment was appreciated by performing incubations with denatured enzymes (Fig. 6a, h). Incubation experiments show that both GlgE activities possess either an α-1,4-glucanotransferase or maltosyltransferase activities depending on the presence of M1P. When M1P is omitted, GlgE activities harbor an α-1,4-glucanotransferase activity exclusively with glucans composed of six or seven glucose units (DP6 or DP7). Interestingly, after 1 h or overnight incubation, DP6 or DP7 are disproportionated with one or two maltosyl moieties leading to the release of shorter (DPn-2) and longer glucans (DPn + 2) (Fig. 6g, n and Supplementary Figs. 4 and 5). The limited number of transfer reactions emphasizes probably a side reaction of GlgE activities. The α-glucanotransferase activity can be also appreciated on native-PAGE containing glycogen. Chain length modification of external glucan chains of glycogen results in increase of iodine interactions visualized as a brownish activity band (Supplementary Fig. 6a). After 1 h of incubation (Supplementary Figs. 4 and 5), both GlgE activities enable the transfer the maltosyl moiety of M1P onto the glucan primer with a DP ≥ 3 (Fig. 6d–f, k–m and Supplementary Figs. 4 and 5). Interestingly, for a longer period of incubation time, both GlgE activities behave either like processive or

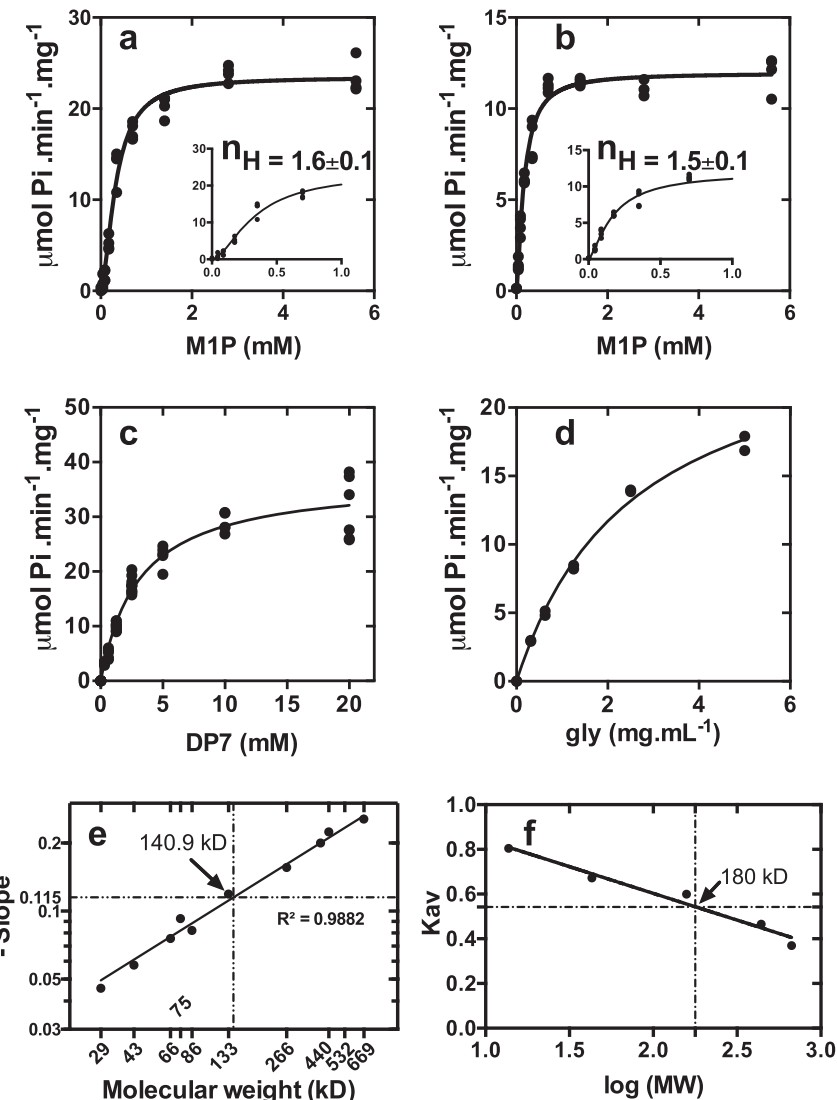

**Fig. 5 Kinetic parameters of recombinant GlgE-EL.** GlgE activity was assayed spectrophotometrically by monitoring the release of inorganic orthophosphate (Pi). Data are presented as individual data points of $n$ independent experiments. M1P saturation plots for GlgE-EL were determined in the presence of 10 mM of maltoheptaose (DP7) ($n = 3$) (**a**) or 10 mg.mL$^{-1}$ of glycogen ($n = 3$) (**b**). At low M1P concentrations (panels), GlgE-EL activity behaves as allosteric enzyme with Hill coefficients ($n_H$) of 1.6 and 1.5, respectively (fit shown as the solid line giving $r^2 = 0.98$). The $S_{0.5}$ (M1P) values for GlgE-EL were determined at 0.33 ± 0.02 mM and 0.16 ± 0.01 mM in the presence of DP7 and glycogen, respectively. In the presence of 2 mM M1P, both DP7 ($n > 3$) (**c**) and glycogen ($n = 2$) (**d**) saturation plots are conformed to the Michaelis–Menten behavior ($n_H$ close to 1) with $K_m$ values of 3.1 ± 0.2 mM and 2.5 ± 0.2 mg.mL$^{-1}$, respectively. The apparent molecular weight of GlgE-EL was determined by native-PAGE (**e**) and size exclusion chromatography (Superose 6 Increase GL 10/300) (**f**) at 140.9 and 180 kDa, respectively, suggesting a dimer of GlgE (76 kD).

distributive enzymes depending on the initial degree of polymerization of the glucan primer. The processive behavior of GlgE enzymes was unexpected since GlgE activity has been reported to operate a double displacement reaction (i.e., Ping-Pong mechanism) involving the release of (2 + n) glucan prior to the next reaction[18]. As depicted in Fig. 6, the synthesis of very long glucan chains, up to 32 glucose residues suggests that both GlgE-EL and GlgE-WC undergo processive-like elongation activities in the presence of maltose (DP2) or maltotriose (DP3). In contrast, when both GlgE activities are incubated in presence of glucan primers with DP ≥ 4, the latter add and immediately release a glucan primer (DP) with an increment of two glucose moieties (DPn + 2) that leads to a distributive elongation behavior. The mechanism underlying the switch between processive-like and distributive elongation activities reflects, probably, a competition of glucan primers for the glucan-binding site in the vicinity of the catalytic domain. Thus,

we can hypothesize that the low affinity of short glucan primers (DP < 4) for glucan-binding sites favors probably iterative transferase reactions onto the same acceptor glucan (i.e., processive-like mode) resulting in the synthesis of long glucan chains whereas glucan primers with DP ≥ 4 compete strongly for the binding site leading to a distributive mode. The discrepancy between GlgE-EL and GlgE-WC to synthesize long glucan chains in the absence (Fig. 6b, i) or in the presence of glucose (DP1) (Fig. 6c, j) might be explained by a higher amount of free maltose observed in denatured GlgE-WC samples (Fig. 6a) by comparison to denatured GlgE-EL samples (Fig. 6h). Despite having taken all precautions (same M1P preparation, buffer pH 7), spontaneous dephosphorylation of M1P occurred more significantly in GlgE-WC samples. We therefore conclude that initial traces of maltose in GlgE-WC samples facilitate the synthesis of long glucan chains in the absence (Fig. 6b) or in the presence of glucose (DP1) (Fig. 6c). To test this hypothesis, crude extract (CE) and purified

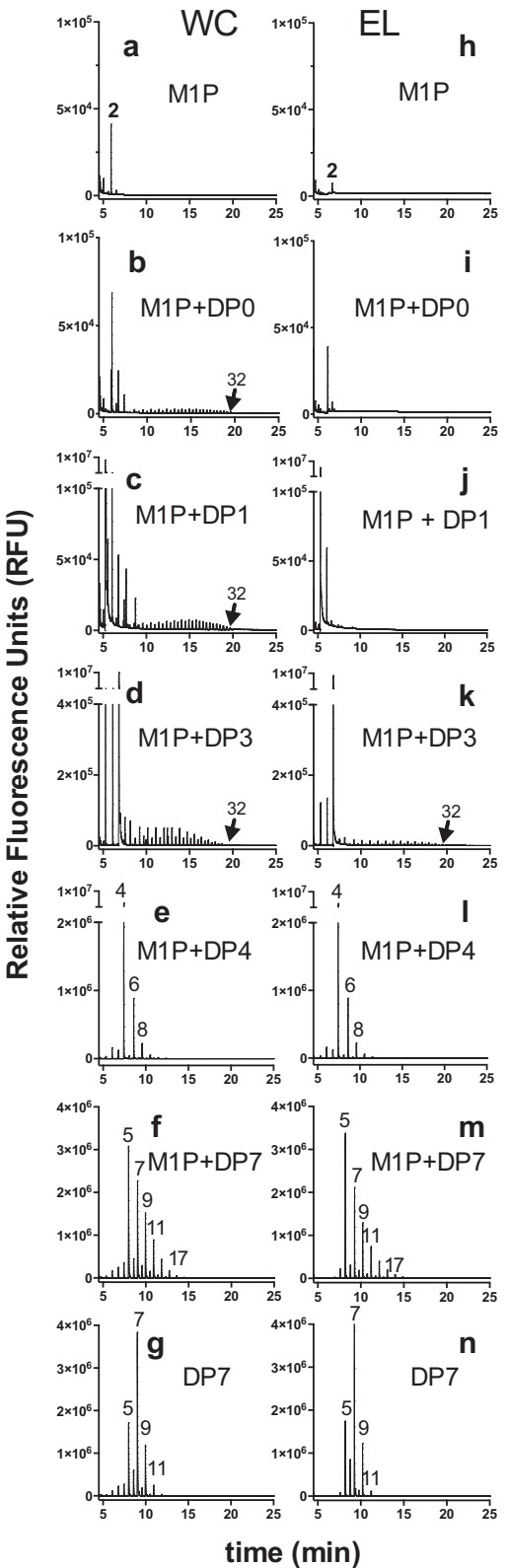

**Fig. 6 FACE analyses of enzymatic reaction products of GlgE activity of *W. chondrophila*, WC and *E. lausannensis*, EL. a, h** Spontaneous dephosphorylation of M1P during overnight incubation was estimated by incubating denatured GlgE enzymes in buffer containing 1.6 mM M1P. The transfer of maltosyl moieties from M1P at 1.6 mM onto non-reducing ends of glucan acceptors (5 mM) were determined in absence of glucan acceptor, DP0 (**b, i**) or in presence of glucose, DP1 (**c, j**), maltotriose, DP3 (**d, k**), maltotetraose, DP4 (**e, l**), and maltoheptaose, DP7 (**f, m**). **g, n** α-1,4-glucanotransferase activities of GlgE were determined by incubating 5 mM of maltoheptaose, DP7, without maltose-1-phosphate. Numbers on the top of fluorescence peaks represent the degree of polymerization of glucan chains.

through interaction with iodine molecules) is detected by soaking the gel in iodine solution. As depicted in Fig. 7a, the synthesis of glucan chains catalyzed by GlgE-EL appears exclusively as dark-blue activity bands inside native-PAGE incubated with 2 mM M1P and not in the absence of M1P.

Altogether, these results suggest that GlgE activities are able to synthesize de novo a sufficient amount of long linear glucans from maltose-1-phosphate. We cannot exclude the role of maltose in the initiation process of glucan synthesis as glucan acceptor since spontaneous dephosphorylation of M1P is unavoidable. We further carried out a series of experiments that consisted to synthesize in vitro high molecular branched glucans by incubating both recombinant glycogen branching enzyme of *W. chondrophila* (GlgB-WC: Supplementary Fig. 6b) and GlgE-EL in the presence of M1P. After overnight incubation, the appearance of α-1,6 linkages or branching points were directly measured by subjecting incubation product on proton-NMR analysis (Fig. 7b). In comparison with M1P and glycogen as controls, proton-NMR spectrum of incubation products shows a typical profile of glycogen-like with signals at 5.6 and 4.9 ppm of proton involved in α-1,4 and α-1,6 linkages. This branched polysaccharide material was further purified and incubated with a commercial isoamylase type debranching enzyme (Megazyme) that cleaves off α-1,6 linkages or branching points. Released linear glucan chains were labeled with APTS and separated according to the degree of polymerization by capillary electrophoresis. The chain length distribution (CLD) of synthesized polysaccharides (Fig. 7c) was compared with glycogen from bovine liver (Fig. 7e). As control, the amounts of free linear glucans were estimated by analyzing the APTS-labeled samples not incubated with commercial debranching enzyme (Fig. 7d, f). In absence of notable amount of free glucan chains (Fig. 7d), the in vitro synthesized polysaccharide harbors a typical CLD similar to animal glycogen with monomodal distribution and maltohexaose (DP6) as most abundant glucan chains. Altogether, these results confirm that GlgE activities display an in vitro function similar to that of glycogen synthase (GlgA) for initiating and elongating the growing glycogen particles.

**Expression of bifunctional TreS-Mak of *Estrella lausannensis*.** To our knowledge, the characterization of the bifunctional TreS-Mak activity has not yet been reported in the literature. The his-tagged TreS-Mak protein purified on nickel columns displays a molecular weight of 115 kDa on SDS-PAGE (Supplementary Fig. 7a) while in solution recombinant TreS-Mak formed a homodimer with an apparent molecular weight of 256 kDa as analyzed by superose 6 column chromatography (Supplementary Fig. 7b). This contrasts with the hetero-octameric complex composed of four subunits of TreS and four subunits of Mak (≈490 kDa) observed in *Mycobacterium smegmatis* in which

GlgE proteins (E1) of *E. lausannensis* were loaded onto nondenaturing polyacrylamide electrophoresis (native-PAGE). After migration, slices of polyacrylamide gel were incubated overnight in buffers containing 0 mM (control) or 2 mM M1P (Fig. 7a). The synthesis of long glucan chains with DP > 15 (minimum number of glucose units required for detection

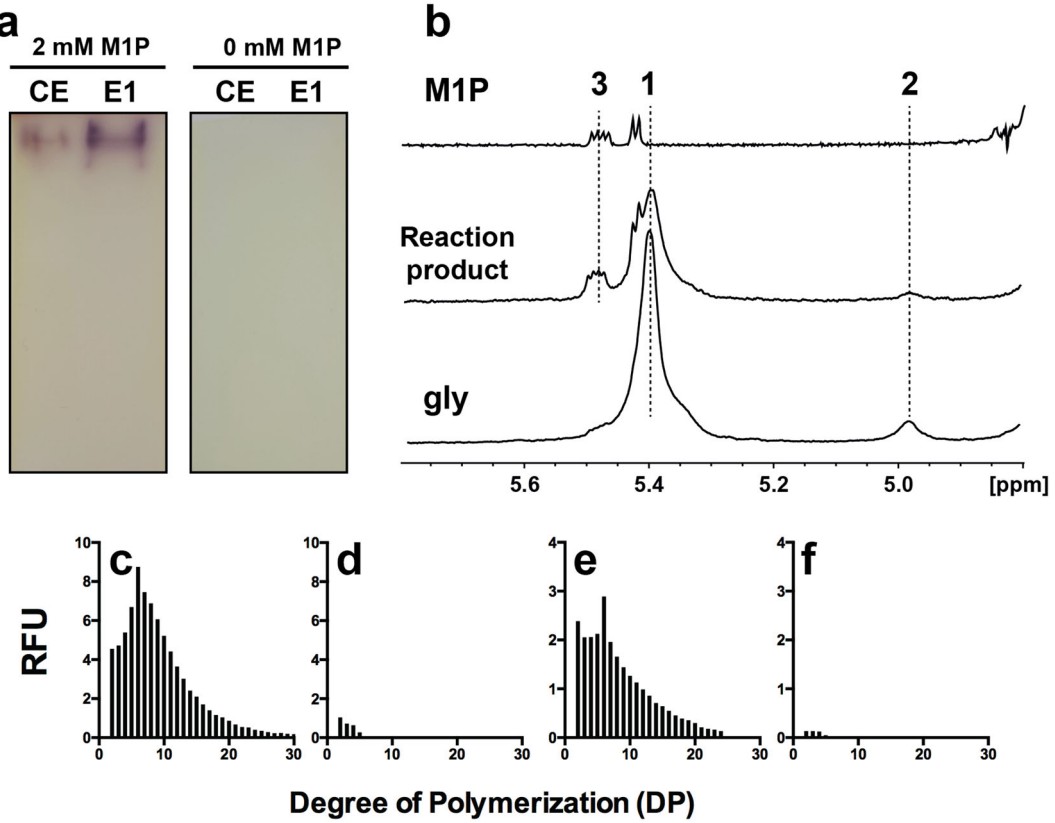

**Fig. 7 De novo synthesis of branched polysaccharides. a** Recombinant GlgE activity of *Estrella lausannensis* from crude extract of *E. coli* (CE) and purified on nickel affinity column (E1) were loaded on nondenaturing polyacrylamide gel. After migration, slices of native-PAGE were incubated in TRIS/acetate buffer containing 2 mM of maltose-1-phosphate (M1P) over 16 h at 25 °C. The synthesis of de novo glucan chains is visualized as dark-blue bands due to the formation of glucan–iodine complexes. Similar in vitro experiments were conducted by adding GlgB activity of *W. chondrophila* to a TRIS/acetate buffer containing GlgE activity and 2 mM of M1P. After overnight incubation, reaction mixture was subjected to $^1$H-NMR analysis. **b** Part of $^1$H-NMR spectra of maltose-1-phosphate (M1P), non-purified reaction mixture, and glycogen (gly) from bovine liver in $D_2O$. Peak #1 (5.45–5.3 ppm) and peak #2 (4.98 ppm) represent the signals of protons involved, respectively, in α-1,4 and α-1,6 linkages while peak #3 (5.47 ppm) represents the characteristic doublet of doublet signals of α-anomeric proton located on C1 of maltose-1-phosphate. The appearances of peak #2 and peak #3 in incubation product indicate the formation of branched polysaccharides composed of α-1,4 and α-1,6 linkages. The presence of peak #3 suggests that M1P was not completely polymerized by GlgE activity of EL. α-Polysaccharides were then purified (see "Methods" for details) and incubated with a commercial isoamylase type debranching enzyme. After overnight incubation, the linear glucan chains released from α-polysaccharides (**c**) and glycogen from bovine liver used as reference (**e**) were separated according to the degree of polymerization by capillary electrophoresis coupled with a fluorescent labeling of reduced-ends. As control, α-polysaccharide (**d**) and glycogen (**f**) samples were directly labeled and analyzed by capillary electrophoresis in order to estimate the content of free-linear glucan chains.

homotetramers of TreS forms a platform to recruit dimers of Mak via specific interaction domain[30,31].

We first confirmed that the N-terminus TreS domain is functional by measuring the interconversion of trehalose into maltose (see "Methods" for details). Previous reports indicated that TreS activities are partially or completely inhibited with 10 mM of divalent cation while a concentration of 1 mM has positive effects. The effect of $Mn^{2+}$ cation on the activity of TreS domain was inferred at 200 mM of trehalose. As depicted in Fig. 8a, the activity of the TreS domain increases only slightly by 1.1-fold from 0 to 1 mM of $Mn^{2+}$ (0.37 μmol maltose. min$^{-1}$.mg$^{-1}$) whereas a noticeable decrease of TreS activity (0.24 μmol maltose. min$^{-1}$.mg$^{-1}$) is obtained at 10 mM of $Mn^{2+}$. As reported in the literature, the TreS activity is also associated with the release of glucose during the interconversion of trehalose into maltose. Because TreS activity is fused with the Mak domain in *E. lausannensis*, we tested the effect of a wide range of concentration of ATP concentration on the interconversion of trehalose (Fig. 8b). Although no revelant effect of ATP was observed on TreS activity at 1 mM (0.43 μmol maltose. min$^{-1}$.mg$^{-1}$), TreS activity decreased by 0.6-fold at 3–10 mM ATP

(0.29 μmol maltose. min$^{-1}$.mg$^{-1}$) and dropped by 2.8-fold when the ATP concentration reaches up to 20 mM (0.15 μmol maltose. min$^{-1}$.mg$^{-1}$). Finally, the apparent $K_m$ value for trehalose was determined at 42.3 ± 2.7 mM in the presence of 1 mM $MnCl_2$ and 0 mM ATP (Fig. 8c). This is consistent with the apparent $K_m$ values for trehalose (50–100 mM) reported in the literature for TreS activity in various species[32]. We further focused on the activity of the maltokinase domain that catalyzes the phosphorylation of maltose in presence of ATP and releases M1P and ADP. The latter was monitored enzymatically via the pyruvate kinase assay in order to express the Mak activity domain as μmol of ADP released.min$^{-1}$. mg$^{-1}$ of protein. The pH and temperature optima were, respectively, determined at 42 °C and pH 8 (Supplementary Fig. 7c, d). Interestingly, the activity of the Mak domain is functional within a wide range of temperature that reflects, probably, the temperature of free-living amoebae or animal hosts. Kinase activities are reported for their requirement in divalent cation in order to stabilize the negatively charged phosphate groups of phosphate donors such as ATP. Therefore, TreS-Mak activity was inferred in the presence of various divalent cations (Fig. 8e). As expected, the

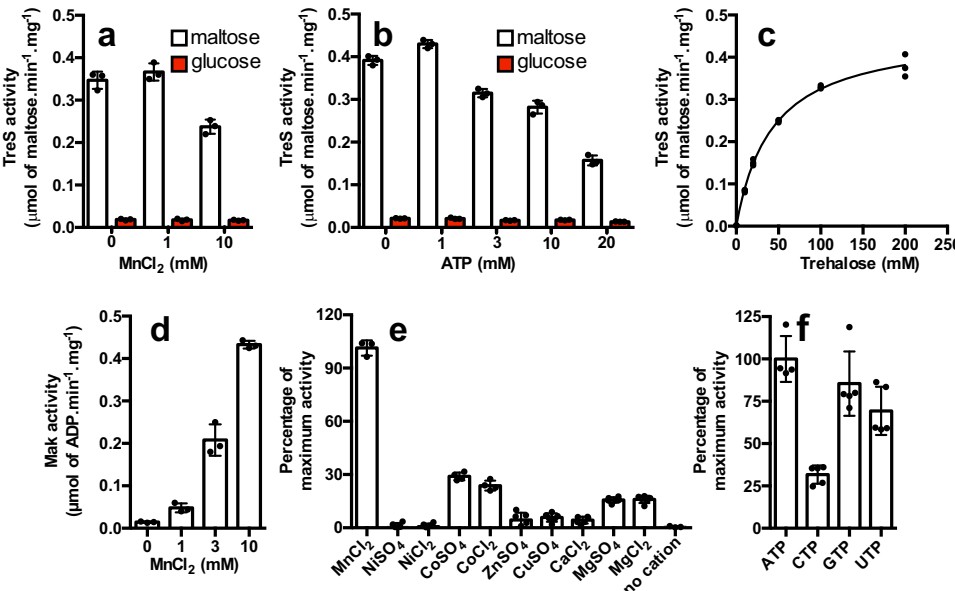

**Fig. 8 Biochemical properties of recombinant TreS-Mak of *Estrella lausannensis*. a** The activity of trehalose synthase domain (TreS) of bifunctional TreS-Mak protein was first conducted at 30 °C pH 8 with 200 mM trehalose in presence of 0, 1, and 10 mM of manganese chloride ($MnCl_2$). The interconversion of trehalose into maltose and subsequent release of glucose (red) were inferred by using amyloglucosidase assay method. The TreS activity is expressed as μmol of maltose/min/mg of protein. **b** The effect of nucleoside triphosphate on TreS activity was determined by measuring the interconversion of trehalose into maltose in the presence of increasing concentration of ATP (0–20 mM) and 200 mM trehalose. **c** The apparent $K_m$ value ($n = 3$) for trehalose was determined in the absence of ATP and 1 mM of $MnCl_2$ by measuring the interconversion of increase concentration of trehalose (0–200 mM) into maltose. **d** Maltokinase activity domain was inferred by measuring the release of ADP during phosphorylation of maltose (20 mM) into M1P in presence of 0, 1, 3, and 10 mM of $MnCl_2$. The Mak activity is expressed as μmol of ADP released/min/mg of protein. **e** The effects of divalent cation $Mn^{2+}$, $Ni^{2+}$, $Co^{2+}$, $Zn^{2+}$, $Cu^{2+}$, $Ca^{2+}$, and $Mg^{2+}$ at 10 mM and **f** ATP, CTP, GTP, and UTP nucleotides on Mak activity were determined and expressed as relative percentage of maximum activity. Data are presented as individual data points with error bar denoting standard deviation of $n \geq 3$ independent experiments.

recombinant TreS-Mak was strictly dependent on divalent cations, in particular, with a noticeable stimulatory effect of $Mn^{2+}$ (Fig. 8d, e). Others tested divalent cations, like $Co^{2+}$, $Mg^{2+}$, $Fe^{2+}$, $Ca^{2+}$, $Cu^{2+}$ activated the Mak activity as well, but to a lower extent, while no effect was observed in presence of $Ni^{2+}$. Interestingly, at variance to Mak activity of *Mycobacterium bovis*, which prefers $Mg^{2+}$, the catalytic site of the Mak activity domain of TreS-Mak binds preferentially $Mn^{2+}$ over $Mg^{2+}$[33], which is consistent with a distinct evolutionary history as depicted in Fig. 2b. Then, nucleotides, ATP, CTP, GTP, and UTP were tested as phosphate donors by measuring the amount of M1P released (Fig. 8f). The data expressed in percentage of activity show that ATP (100%), GTP (85%), UTP (70%), and to a lower extent CTP (31%) are efficient phosphate donors. Altogether, we demonstrated that TreS and Mak domains are functional in the fused protein TreS-Mak of *E. lausannensis*. The reversible interconversion of trehalose combined with an intracellular trehalose concentration probably below 42 mM (the intracellular trehalose concentration was estimated at $40 \pm 10$ mM inside one cell of *E. coli* strain overexpressing OtsA/OtsB[34]) suggest that irreversible phosphorylation of maltose drives the synthesis of M1P.

***Estrella lausannensis* and *Waddlia chondrophila* accumulate glycogen particles within the cytosol of EB via GlgE pathway.** Incubation experiments reported above have shown that branched polysaccharide can be synthesized in vitro in the presence of maltose-1-phosphate and both GlgE and GlgB activities. This prompted us to examine the presence of glycogen particles in thin section of *E. lausannensis* and *W. chondrophila* by transmission electron microscopy. After 24 h post infection with both *Chlamydiales* (Fig. 9a, c), we purified elementary bodies (Fig. 9b, d)

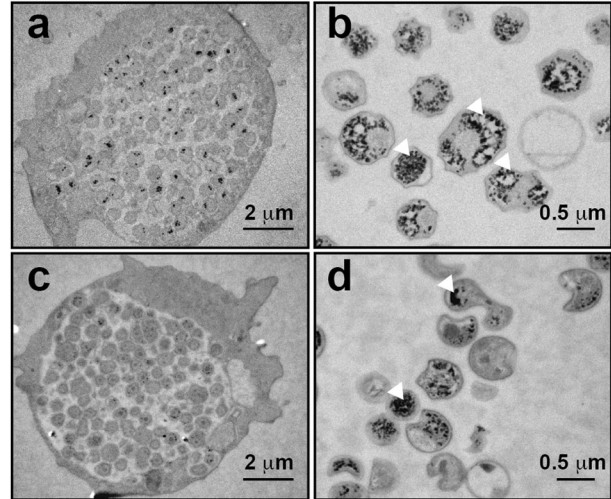

**Fig. 9 Glycogen accumulation in *Estrella lausannensis* and *Waddlia chondrophila*.** Glycogen particles (white head arrows) in *E. lausannensis* (**a**, **b**) and *W. chondrophila* (**c**, **d**) were observed by TEM after periodic acid thiocarbohydrazide-silver proteinate staining of ultrathin sections of 24 h post infected *A. castellanii* with *E. lausannensis* (**a**) and *W. chondrophila* (**c**) or purified bacteria (**b**, **d**).

from the infected *Acanthamoeba castellanii* cultures and subjected them to specific glycogen staining based on the periodic acid method, which is considered to be one of the most reliable and specific methods for staining glycogen[35]. Glycogen particles appear as electron-dense particles (white head arrows) in the cytosol of elementary bodies of *E. lausannensis* and

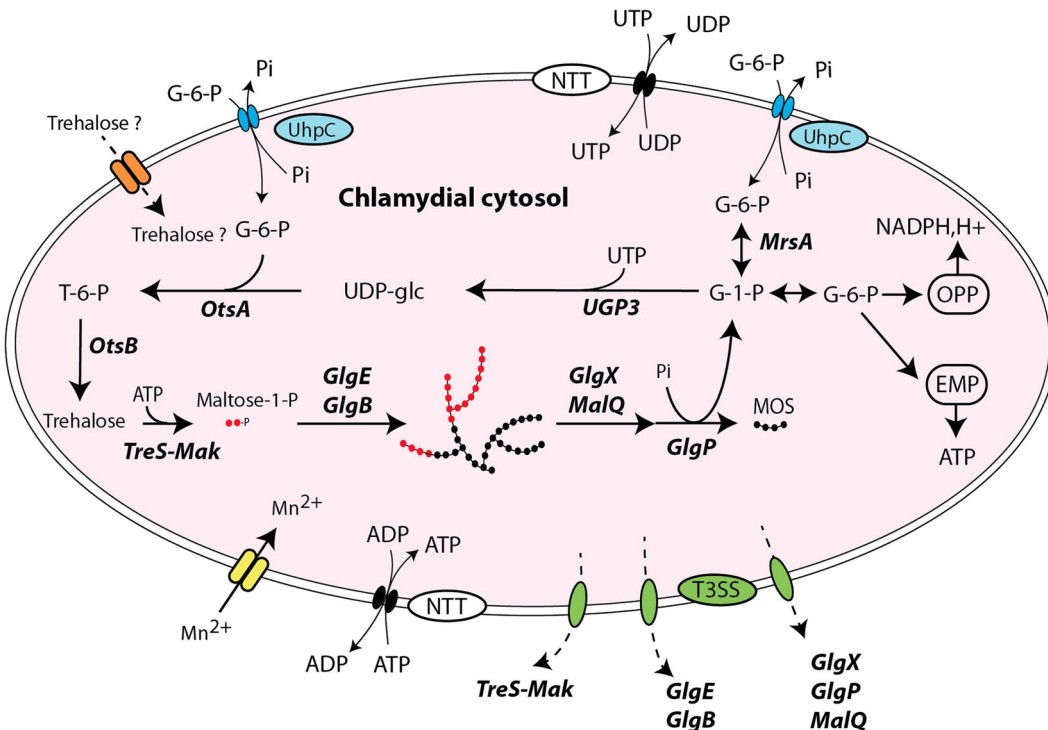

**Fig. 10 Glycogen metabolism network in *Waddliaceae* and *Criblamydiaceae* families.** Glucose-6-phosphate (G-6-P) and UTP/ATP are transported in the cytosol via Uhpc and NTT translocators. The first committed step consists of isomerization of G-6-P into glucose-1-phosphate (G-1-P) catalyzed by glucose-6-phosphate isomerase activity (MrsA). UDP-glucose pyrophosphorylase (UGP3) synthesizes UDP-glucose from G-1-P and UTP. Both trehalose-6-phosphate synthase (OtsA) and trehalose-6-phosphate phosphatase (OtsB) convert nucleotide-sugar and G-6-P into trehalose. The bifunctional TreS-Mak activity supplies the maltosyltransferase activity (GlgE) in maltose-1-phosphate (M1P). De novo glucan initiation and elongation properties of GlgE and branching enzyme activity (GlgB) allow the appearance of α-polysaccharide (i.e., glycogen) made of α-1,4 and α-1,6 linkages. The synergic action of glycogen phosphorylase (GlgP), debranching enzyme (GlgX), and α-1,4-glucanotransferase (MalQ) depolymerize glycogen into G-1-P and short malto-oligosaccharides (MOS). The former fuels both oxidative pentose phosphate (OPP) and Embden–Meyerhof–Parnas (EMP) pathways that supply the extracellular forms (elementary bodies) in reduced power (NADPH,H+) and ATP, respectively. Divalent cations $Mn^{2+}$ required for TreS-Mak activity are probably imported via ABC transporter composed of MntA, MntB, and MntC 3 subunits identified in chlamydial genomes. *Waddliaceae* and *Criblamydiaceae* may manipulate the carbon pool of the host by uptaking trehalose through a putative disaccharide transporter (orange/dash arrow) or by secreting glycogen-metabolizing enzymes through type three-secretion system (green/dash arrow).

*W. chondrophila.* Interestingly, because *Waddlia chondrophila* infects animal cells, which do not synthesize trehalose, this suggests that trehalose must be synthesized by the bacteria itself[36]. Based on the five different trehalose pathways described in prokaryotes (for review, see ref. [37]), we found that trehalose biosynthesis is limited to so-called "environmental *Chlamydiae*" and is not present in the *Chlamydiaceae* family. Among chlamydial strains with GlgE pathway, *P. phocaeensis* and *P. neagleriophila* synthesize trehalose through the TreY-TreZ pathway while the OtsA-OtsB pathway was found in both *E. lausannensis* and *W. chondrophila*. Importantly, *otsA* and *otsB* genes encode trehalose-6-phosphate synthase and trehalose-6-phosphate phosphatase, respectively. OtsA activity condenses glucose-1-phosphate and UDP-glucose into trehalose-6-phosphate. However, BLAST search did not evidence the classical *galU* gene encoding UDP-glucose pyrophosphorylase, which synthesizes UDP-glucose from glucose-1-phosphate and UTP in both *E. lausannensis* and *W. chondrophila*, but rather a non-GalU type UDP-glucose pyrophosphorylase homolog to UGP3 of plants[38]. Based on this work and taking into account the current genome analysis, we propose that glycogen metabolism pathway in *W. chondrophila* and *E. lausannensis* occur as depicted in Fig. 10.

## Discussion

The present study examined the glycogen metabolism pathway in *Chlamydiae* phylum. Unlike other obligate intracellular bacteria,

*Chlamydiae* have been documented to retain their capacity to synthesize and degrade the storage polysaccharide with the notable exception of the *Criblamydiaceae* and *Waddliaceae* families, for which the key enzyme of glycogen biosynthesis pathway, ADP-glucose pyrophosphorylase activity was reported missing[12,39]. All mutants deficient in GlgC activity are associated with glycogen-less phenotypes and so far no homologous gene encoding a GlgC-like activity has been established among prokaryotes[40,41]. To our knowledge, only two cases have been documented for which the GlgC activity has been bypassed in the classical GlgC pathway. The ruminal bacterium *Prevotella bryantii* that does not encode an ADP-glucose pyrophosphorylase (*glgC*) gene has replaced the endogenous *glgA* gene with an eukaryotic UDP-glucose-dependent glycogen synthase[42,43]. The second case reported concerns the GlgA activity of *Chlamydia trachomatis* which has evolved to polymerize either UDP-glucose from the host or ADP-glucose produced by GlgC activity into glucose chains[44]. In order to get some insight into *Criblamydiaceae* and *Waddliaceae* families, a survey of glycogen-metabolizing enzymes involved in the classical GlgC pathway and in the recently described GlgE pathway was carried out over 47 chlamydial species representing the diversity of chlamydiae phylum. As expected, we found a complete GlgC pathway in most chlamydial families, and the most astonishing finding was the occurrence of GlgE pathway in three phylogenetically related *Parachlamydiaceae*, *Waddliaceae*, and *Criblamydiaceae* families.

Our genomic analysis also pinpointed a systematic lack of *glgC* gene in 6 draft genomes of candidatus *Enkichlamydia* sp. Those genomes are derived from metagenomic studies and were estimated to be 71–97% complete. If we assume the loss of *glgC*, the characterization of glycogen synthase with respect to its ability to use various nucleotide sugars should shed light on the glycogen pathway in this case and as a result would possibly provide another example of the bypassing of GlgC activity in the classical GlgC pathway. In addition to the occurrence of GlgE pathway, a detailed genomic analysis of *Waddliaceae* and *Criblamydiaceae* families has revealed a large rearrangement of *glg* genes of the GlgC pathway, which had led to the loss of the *glgP* gene and a fusion of *glgA* and *glgB* genes. This fusion appears exceptional in all three domains of life and no other such examples have been reported. In *E. lausannensis* (*Criblamydiaceae* fam.), the fusion of *glgA-glgB* genes is associated with a non-sense mutation resulting in premature stop codon in the open reading frame of the GlgA domain precluding the presence of the fused branching enzyme. We have shown that the glycogen synthase domain of chimeric GlgA–GlgB of *W. chondrophila* was active and remained ADP-glucose dependent while its branching enzyme domain already appears to be non-functional due to the presence of the GlgA domain at the N-terminal extremity that prevents the branching enzyme activity. In line with these observations and at variance with *Parachlamydiaceae* which have only maintained the *glgB* gene of the GlgC pathway, both *Waddliaceae* and *Criblamydiaceae* have selected and conserved the *glgB2* gene from the incoming GlgE operon, thereby, further suggesting that the glycogen branching activity domain is indeed defective or impaired in all GlgA–GlgB fusions. Overall, this study clearly implies that the GlgC pathway does not operate in both the *Waddliaceae* and *Criblamydiaceae* families. In addition, it appears possible that the genes required for the presence of a functional GlgC pathway are at different stages of disappearance from these genomes, as suggested by the non-sense mutation in glgA–glgB gene of *E. lausannensis*[45].

We further investigated the GlgE glycogen biosynthesis pathway in *Chlamydiales*. A series of biochemical characterizations have shown that GlgE activities are capable of transferring maltosyl residue of maltose-1-phosphate onto linear chain of glucose. More remarkably, GlgE activities fulfill the priming function of glycogen biosynthesis as described for the GlgA activity in the GlgC pathway[10]. We have shown that the GlgE activities switch between the processive-like or distributive modes of polymerization depending on the initial presence of glucan chains. Thus the "processive mode" of GlgE activity yields long glucan chains (DP > 32) and is favored in their absence or in the presence of short glucan primers (DP < 4). This "processive mode" of GlgE activity fills up the critical function of initiating long glucan chains that will be taken in charge by the branching enzyme in order to initiate the formation of glycogen particles. In vitro incubation experiments performed in the presence of M1P and/or branching enzyme activity further confirmed that GlgE activity is by itself sufficient for synthesizing de novo a branched polysaccharide with high molecular weight. At variance with mycobacteria and *Streptomycetes*, trehalose synthase (TreS) and maltokinase (Mak) activities of *Chlamydiales* form a bifunctional enzyme composed of TreS and Mak domains at the N- and C-terminus, respectively, which has never been reported to our knowledge. The fused TreS-Mak activity is functional and mediates the trehalose conversion into maltose and the phosphorylation of maltose into maltose-1-phosphate in the presence of ATP, GTP, or UTP as phosphate donors. In contrast to mycobacteria, the maltose kinase domain requires preferentially manganese rather magnesium as divalent cation[33].

The fact that the occurrence of GlgE pathway is limited to a few chlamydial families has led us to wonder about the origin of this operon. Our phylogeny analyses suggest that GlgE operons identified in chlamydia species share a common origin but are only distantly related to the GlgE operon from Actinobacteria (i.e., Mycobacteria). We could not determine whether the presence of the GlgE pathway predated the diversification of chlamydiae or whether the operon was acquired by lateral gene transfer by the common ancestor of the *Criblamydiaceae*, *Waddliaceae*, and *Parachlamydiaceae* from another member of PVC superphylum. One fair inference is that the genome of the common ancestor of *Waddliaceae*, *Criblamydiaceae*, and *Parachlamydiaceae* families encoded both GlgC- and GlgE-pathways. The loss of both glgP and glgC and fusion of glgA and glgB occurred before the emergence of *Waddliaceae* and *Criblamydiaceae* and may involve one single-deletion event if we presume a glA/glgC/glgP/glgB gene arrangement in the common ancestor. While GlgE pathway was maintained in *Waddliaceae* and *Criblamydiaceae* due to the mandatory function of glycogen in *Chlamydiales*, most of the members of *Parachlamydiaceae* retained only the GlgC pathway except for two *Protochlamydia* species. The redundancy of glycogen metabolism pathway in *P. naegleriophila* species is quite surprising and goes against the general rule of genome optimization of obligate intracellular bacteria. It is worthy to note that the *P. naegleriophila* species was originally isolated from a protist *Naegleria* sp. *N. fowleri*, the etiological agent of deadly amoebic encephalitis in humans, stores carbon exclusively in the form of trehalose and is completely devoid of glycogen-synthesis genes[46]. Therefore, it is tempting to hypothesize that *P. neagleriophila* use the retained GlgE pathway to effectively mine the trehalose source of its host either by uptaking trehalose from its host through a putative disaccharide transporter or by secreting via type three-secretion system enzymes of GlgE pathway. Our preliminary experiments based on heterologous secretion assay in *Shigella flexneri* suggested that GlgE and TreS-Mak could be secreted by the type three-secretion system (Supplementary Fig. 8). Like *Chlamydiaceae*, the secretion of enzymes chlamydial glycogen metabolism pathway might be a strategy for manipulating the carbon pool of the host[44]. As reported for *P. amoebophila* with respect to D-glucose[39], the utpake of host's trehalose provides an important advantage in terms of energy costs. In comparison with the GlgC pathway, one molecule of ATP is required to incorporate two glucose residues onto growing polysaccharide; at the scale of one glycogen particle synthesis this may represent a relevant amount of ATP saving. The uptake of radiolabeled trehalose by host-free elementary bodies may or not support this hypothesis.

The remarkable and unique preservation of glycogen metabolism among otherwise glycogen-less intracellular bacteria through the bottleneck of genome reduction process suggests an unexpected function of glycogen that has been hitherto underestimated within Chlamydiae. As a result, the question arises as to why chlamydiae have maintained glycogen metabolism pathway, making them unique among obligate intracellular bacteria. It is worthy to note that most of obligate intracellular bacteria *Anaplasma* spp., *Ehrlichia* spp., *Wolbachia* spp., and *Rickettsia* spp. do not experience environmental stresses like Chlamydiae and *Coxellia burnetii*[47]. They thrive in nutrient rich environments either in animal or insect hosts. Losses of metabolic functions such as carbon storage metabolism in obligate intracellular bacteria are balanced by the expression of a wide variety of transporters for the uptake metabolites from the host. Except for ultra-resistant spore-like forms of *C. burnetti* named small cell variants, over the last decade, our perception of EB has switched from an inert spore-like form to metabolic active form capable of transcription and translation activities[39,48]. The combination of

different "omic" approaches performed on purified RB and EB of *C. trachomatis* and *P. amoebophila* have shown that genes involved in glycogen and energy metabolism pathways are upregulated in the late stage of development[4,49,50] and most remarkably, the uptake of glucose and glucose-6-phosphate by EBs of *P. amoebophila* and *C. trachomatis* improves significantly the period of infectivity[39,51]. Accordingly, it seems reasonable to argue that the primary function of cytosolic glycogen in EBs is to fuel metabolic processes (i.e., glycolysis, pentose phosphate) when EBs are facing up the poor nutrient environment (Fig. 10). Future investigations should provide new opportunity to delineate the function of glycogen in chlamydiae especially with the development of forward genetic approaches[52,53]. Finally, the use of GlgE inhibitors initially designed against mycobacterial infections and to some extent the use of inhibitors of chlamydial glycogen-metabolizing enzymes might define new attractive drugs to treat *W. chondrophila*, since this *Chlamydia*-related bacteria has been increasingly recognized as a human pathogen[54,55].

## Methods

**Comparative genomic analysis of glycogen metabolic pathways**. In order to gain insight into Chlamydiae's glycogen metabolism, homologs of proteins part of the glycogen pathway of *E. coli* and of *M. tuberculosis* were searched with BLASTp in 220 genomes and metagenome-derived genomes from 47 different chlamydial species available on the ChlamDB database (https://chlamdb.ch/, https://academic.oup.com/nar/article/48/D1/D526/5609527)[56]. The completeness of metagenome-derived and draft genomes was estimated with checkM based on the identification of 104 nearly universal bacterial marker genes[3]. The species phylogeny has also been retrieved from ChlamDB website.

**Microscopy analysis**. Fresh cultures of *Acanthamoeba castellanii* grown in 10 mL YPG (Yeast extract, peptone, glucose) were infected with 1-week-old 5-μm-filtered suspension of *E. lausannensis* or *W. chondrophila* ($10^5$ cells.mL$^{-1}$), as previously reported[57]. Samples of time-course infection experiments were harvested at 0, 7, 16, and 24 h post infection by centrifuging the infected *A. castellanii* cultures at $116 \times g$. Pellets were then fixed with 1 mL of 3% glutaraldehyde for at 4 °C, followed by three washing steps with 100 mM PBS. Cells were further fixed with 1% osmium tetroxide in PBS for 1 h at room temperature. Samples were dehydrated with subsequent increasing ethanol washes (50–100%). Samples were then transferred into propylene oxide and incubated overnight in an epoxy resin (Epon) mixed with 50% propylene oxide[58]. Thin sections (70-nm thick) were cut with a diamond knife in a Leica UC6 microtome and poststained with periodic acid thiosemicarbazide silver proteinate. All specimens were observed with a Philips CM200 transmission electron microscope operating at 80 kV. Images were recorded on Kodak SO163 film.

**Glycogen synthase and glycogen branching enzyme activities**. Fused *glgAglgB* genes of *E. lausannensis* and *W. chondrophila* were amplified using pairs of primers harboring attB sites as described in Supplementary Table 2. PCR products were then cloned in the pET15b (Novagen) plasmid modified compatible with Gateway™ cloning strategy. The expression of his-tagged recombinant protein GlgA–GlgB was performed in the derivative BW25113 strain impaired in the endogenous glycogen synthase activity (ΔglgA). Recombinant proteins were separated by nondenaturing PAGE containing 0.6% rabbit glycogen (Sigma-Aldrich). After electrophoresis, gels were incubated overnight at room temperature in glycogen synthase buffer containing either 1.2 mM ADP-Glc or 1.2 mM UDP-Glc (70 mM Gly-Gly, pH 7.5, 135 mM ($NH_4$)$_2$SO$_4$, 280 mM NaF, 330 mM trisodium citrate, 290 mM sodium acetate, and 67 mM β-mercaptoethanol). Glycogen synthase activities were then visualized as dark activity bands after soaking native-PAGE gels in iodine solution (0.5 g of I2 and 10 g of KI)[59]. The nucleotide-sugar specificity of glycogen synthase was carried out by following the incorporation of $^{14}$C-Glc of radiolabelled ADP-$^{14}$C-[U]-glc or UDP-$^{14}$C-[U]-glc into glycogen particles during 1 h at 30 °C. The reaction was stopped by precipitating labeled glycogen with 1 mL of 75% [v/v] and 1% [w/v] methanol-KCl. The samples were stored at −20 °C for 10 min and then centrifuged for 5 min at $3000 \times g$ at 4 °C. After centrifugation, the glycogen pellets were suspended with 200 μL of distilled water. This step was repeated twice before mixing the sample with 2.5 mL of scintillation liquid. The radioactivity incorporated into glycogen was determined by liquid scintillation counting. Radiolabelled nucleotide sugars were purchased from Amersham-Biosciences.

**Phylogeny analysis**. Homologous sequences of TreS-Mak and GlgE were carried out by BLAST against the nr database from NCBI with, respectively, WP_098038072.1 and WP_098038073.1 sequences of *Estrella lausannensis*. We retrieved the top 2000 homologs with an *E*-value cut off ≤10$^{-5}$ and aligned them using MAFFT[60] with the fast alignment settings. Block selection was then performed using BMGE[61] with a block size of four and the BLOSUM30 similarity matrix. Preliminary trees were generated using Fasttree[62] and 'dereplication' was applied to robustly supported monophyletic clades using TreeTrimmer[63] in order to reduce sequence redundancy. For each protein, the final set of sequences was selected manually. Proteins were realigned with MUSCLE[64] and block selection was carried out using BMGE with a block size of four and the matrix BLOSUM30. Phylogenetic trees were inferred using Phylobayes[65] under the catfix C20 + Poisson model with the two chains stopped when convergence was reached (maxdiff < 0.1) after at least 500 cycles, discarding 100 burn-in trees. Bootstrap support values were estimated from 100 replicates using IQ-TREE[66] under the LG4X model and mapped onto the Bayesian tree.

**GlgE and TreS-Mak expressions**. *glgE* and *treS-mak* genes were amplified from the genomic DNA of *E. lausannensis* and *W. chondrophila* using the following pairs of primers F_glgE_EL/R_glgE_EL, F_glgE_WC/R_glgE_WC, and F_treS-mak_EL/R_treS-mak_EL (Supplementary Table 1). The sequences and uniprot accession numbers are displayed in the Supplementary Figs. 2e and 7g. The PCR products were cloned in the expression vector pET15b (Novagene) or VCC1 (P15A replicon). The resulting plasmids pET-GlgE-WC, pET-GlgE-EL, and VCC1-treS-mak-EL were transferred to *E. coli* Rosetta™ (DE3; pRARE) or BL21-AI™. The first transformation experiment in Rosetta™ (DE3) *E. coli* strain with VCC1-tres-mak-EL plasmid did not yield colonies in spite of the absence of inducer (IPTG) (Supplementary Fig. 7e). We presumed that a basal transcription of *treS-mak* gene associated with a substantial intracellular amount of trehalose (estimated at 8.5 mM in *E. coli* cell spread on Luria-Broth agar medium[34]) lead to the synthesis of highly toxic maltose-1-phosphate. This encouraging result prompted us to perform expression of TreS-Mak protein in BL21-AI strain. The expression of his-tagged proteins was induced in Lysogeny Broth (LB) or Terrific Broth (TB) with 0.5 mM isopropyl β-D-1-thiogalactopyranoside (IPTG) or 1 mM IPTG/0.2% L-arabinose at the mid-logarithmic phase growth ($A_{600} = 0.5$) or by using auto-inducible medium as described in Fox and Blommel[67]. After 18 h of incubation at 30 °C, cells were harvested at $4000 \times g$ at 4 °C during 15 min. Cell pellets were stored at −80 °C until purification step on Ni$^{2+}$ affinity column.

**Protein purification**. Cell pellets from 100 mL of culture medium were suspended in 1.5 mL of cold buffer (25 mM Tris/acetate pH 7.5). After sonication (three times 30 s), proteins were purified on Ni$^{2+}$ affinity column (Roth) equilibrated with washing buffer (300 mM NaCl, 50 mM sodium acetate, 60 mM imidazole, pH 7) and eluted with a similar buffer containing 250 mM imidazole. Purification steps were followed by SDS-PAGE and purified enzymes quantified by Bradford method (Bio-Rad).

**Evidence of GlgE-like activity by thin-layer chromatography**. Maltosyltransferase activities of GlgE proteins of *E. lausannensis* and *W. chondrophila* were first evidenced by incubating the purified recombinant enzymes overnight at 30 °C with 10 mg.mL$^{-1}$ glycogen from rabbit liver (Sigma-Aldrich) and 20 mM orthophosphate in 20 mM Tris-HCl buffer (pH 6.8). The reaction products were separated on thin-layer chromatography Silica gel 60 W (Merck) using the solvent system butanol/ethanol/water (5/4/3 v/v/v) before spraying orcinol (0.2%)-sulfuric (20%) solution to visualize carbohydrates.

**Maltose-1-phosphate purification**. Maltose-1-phosphate (M1P) was enzymatically produced from 20 mL of reaction mixture containing 1 mg of GlgE-EL, 10 mg.mL$^{-1}$ of potato amylopectin and 20 mM of orthophosphate in 20 mM Tris/HCl pH 6.8. After overnight incubation at 30 °C, M1P and over left polysaccharide were separated according to their molecular weight on size exclusion chromatography (TSK-HW 50 (Toyopearl), column size: length 48 cm, diameter 2.3 cm) equilibrated with 1% ammonium acetate at the flow rate of 1 mL.min$^{-1}$. In the second step, M1P and orthophosphate were separated by using anion exchange chromatography (Dowex 1X8 100–200 mesh acetate form (Bio-Rad), column size: length 28 cm, diameter 1.6 cm) equilibrated with 0.5 M potassium acetate pH 5 at the flow rate of 0.75 mL.min$^{-1}$. M1P containing fractions were then neutralized with an ammoniac solution (30%). Finally, M1P was desalted using cation exchange chromatography (Dowex 50WX8 50–100 mesh H$^+$ form (Bio-Rad), column size: length 10 cm, diameter 1 cm) equilibrated with water. Around 10 mg of maltose-1-phosphate were recovered from a reaction mixture of 20 mL, with a yield of ~5%. Maltose-1-phosphate was also produced by incubating overnight at 30 °C recombinant TreS-Mak protein from *E. lausannensis* with 20 mM ATP, 20 mM maltose, 10 mM MnCl$_2$, 125 mM imidazole, and 150 mM NaCl. Following an anion exchange chromatography step as described above, maltose-1-phosphate was purified from remaining maltose and salts using Membra-cell MC30 dialysis membrane against ultrapure water. This purification procedure leads to a better yield (around 8%). Maltose-1-phosphate was further characterized by mass spectrometry and proton-NMR analysis.

**Proton-NMR analysis of maltose-1-phosphate**. Sample was solubilized in D$_2$O and placed into a 5-mm tube. Spectra were recorded on 9.4 T spectrometer ($^1$H resonated at 400.33 MHz and $^{31}$P at 162.10 MHz) at 300 K with a 5 mm TXI

probehead. Used sequences were extracted from Bruker library sequence. Delays and pulses were optimized for this sample.

**MALDI-TOF MS analysis**. P-maltose was analyzed by a MALDI-QIT-TOF Shimadzu AXIMA Resonance mass spectrometer (Shimadzu Europe; Manchester, UK) in the positive mode. The sample was suspended in 20 µL of water. In total, 0.5 µL of sample was mixed with 0.5 µL of DHB matrix on a 384-well MALDI plate. DHB matrix solution was prepared by dissolving 10 mg of DHB in 1 mL of a 1:1 solution of water and acetonitrile. The low mode 300 (mass range $m/z$ 250–1300) was used and laser power was set to 100 for 2 shots each in 200 locations per spot.

**Kinetic parameters of GlgE and TreS-Mak of *E. lausannensis***. GlgE activity was monitored quantitatively in the elongation direction by the release of orthophosphate using the Malachite Green Assay Kit (Sigma-Aldrich) following the manufacturers' instructions. The concentration of released free orthophosphate was estimated from a standard curve. The absorbency was monitored at 620 nm with Epoch microplate spectrophotometer (Biotek) after incubating with the malachite green reagent for 30 min. Kinetic parameters of GlgE-EL were determined in triplicates at 30 °C in 15 mM Tris/HCl buffer at pH 6.8 and 0.5 µg.mL$^{-1}$ of GlgE-EL. Enzymatic reactions were stopped at 95 °C for 2 min. Saturation plots for maltose-1-phosphate (0.044, 0.088, 0.175, 0.35, 0.7, 1.4, 2.8, and 5.6 mM) were carried out in the presence of 10 mM of maltoheptaose or 10 mg.mL$^{-1}$ of glycogen from bovine liver (Sigma-Aldrich) whereas 2 mM maltose-1-phosphate were used to get saturation plots for maltoheptaose (0.313, 0.625, 1.25, 2.5, 5, 10, and 20 mM) and glycogen from bovine liver (0.313, 0.625, 1.25, 2.5, and 5 mg.mL$^{-1}$). Kinetic constants were determined by fitting initial velocity curves to Michaelis–Menten ($v = \frac{V_{max}S}{K_m+S}$) and allosteric ($v = \frac{V_{max}S^h}{S_{0.5}^h+S^h}$) models using GraphPad Prism version 5. Optimal temperature and pH were assayed with 1 mM maltose-1-phosphate and 5 mM maltoheptaose, respectively, in 25 mM Tris/HCl pH 6.8 and at 30 °C. Temperature was tested in the range of 15–45 °C and pH between 3 and 8.8 with different buffers: 25 mM sodium acetate at pH 3.7, 4.8, and 5.2; 25 mM sodium citrate at pH 3, 4, 5, and 6; 25 mM Tris/HCl at pH 6.8, 7, 7.5, 7.8, 8, and 8.8.

Our preliminary investigations indicated that imidazole stabilizes or is required for the maltokinase activity domain (Supplementary Fig. 7f). Hence all incubation experiments have been conducted in the presence of 125 mM of imidazole. The trehalose synthase domain (TreS) of TreS-Mak protein was monitored following the interconversion of trehalose into maltose and glucose. The incubation experiments were conducted in the incubation buffer (125 mM imidazole pH 8, 150 mM NaCl) containing various concentrations of trehalose (0–200 mM) or MnCl$_2$ (0–10 mM) and in the presence of 0–10 mM of ATP. After stopping the reaction at 95 °C during 2 min, reaction samples (15 µL) were incubated 30 min at 58 °C with 30 µL of citrate buffer (100 mM sodium citrate pH 4.6) containing either 0.4 U of amyloglucosidase from *Aspergillus niger* (Megazyme) or no amyloglucosidase. The amounts of glucose were then determined after adding 100 µL of NADP/ATP buffer (500 mM triethanolamine hydrochloride pH 7.8, 3.4 mM NADP$^+$, 5 mM MgSO$_4$, 10 mM ATP). The increase of absorbancy at 340 nm was monitored after adding 1.2 U of hexokinase and 0.6 U of glucose-6-phosphate dehydrogenase (Megazyme) using Epoch spectrophotometer. The glucose and maltose were estimated using standard curves.

The maltokinase activity of TreS-Mak protein was monitored following the amount of nucleoside bi-phosphate released. If not stated otherwise, enzymatic reactions were carried out during 30 min at 30 °C in the incubation buffer containing 125 mM imidazole pH 8, 150 mM NaCl, 20 mM maltose, 20 mM ATP, 10 mM MnCl$_2$, and 25 µg mL$^{-1}$ of TreS-Mak_EL. Twenty microliters of reaction mixtures were added to 80 µl of pyruvate kinase buffer (75 mM Tris/HCl pH 8.8, 75 mM KCl, 75 mM MgSO$_4$, 2 mM phosphoenolpyruvate, 0.45 mM NADH). The decrease of absorbancy at 340 nm was monitored after adding 5 U of L-lactic dehydrogenase from rabbit muscle (Sigma-Aldrich) and 4 U of pyruvate kinase from rabbit muscle (Sigma-Aldrich) during 30 min. The amount of nucleoside diphosphate was estimated from a standard curve of ADP.

**Chain length distribution analyses**. GlgE activity was qualitatively monitored using fluorophore-assisted carbohydrates electrophoresis (FACE). Both GlgE-EL (3.5 nmol Pi.min$^{-1}$.mg$^{-1}$) and GlgE-WC (1.4 nmol Pi.min$^{-1}$.mg$^{-1}$) activities were incubated at 30 °C during 1 and 16 h in presence of 5 mM of malto-oligosaccharides (glucose to maltoheptaose) and 1.6 mM of M1P in a final volume of 100 µL. Reactions were stopped at 95 °C for 5 min and supernatants recovered after centrifugation. Samples were dried and solubilized in 2 µL 1 M sodium cyanoborohydride (Sigma-Aldrich) in THF (tetrahydrofurane) and 2 µL 200 mM ATPS (8-aminopyrene-1,3,6-trisulfonic acid trisodium salt, Sigma-Aldrich) in 15% acetic acid (v/v). Samples were then incubated overnight at 42 °C. After addition of 46 µL ultrapure water, samples were again diluted 300 times in ultrapure water prior to injection in a Beckman Coulter PA800-plus Pharmaceutical Analysis System equipped with a laser-induced fluorescence detector. Electrophoresis was performed in a silicon capillary column (inner diameter: 50 µm; outer diameter: 360 µm; length: 60 cm) rinsed and coated with carbohydrate separation gel buffer-N (Beckman Coulter) diluted three times in ultrapure water before injection (7 s at 10 kV). Migration was performed at 10 kV during 1 h.

One milligram glycogen from bovine liver (Sigma) and de novo polysaccharide produced from overnight incubation of 2 mg maltose-1-phosphate with 30 µg GlgE-EL and 200 µg GlgB-WC (sequence and uniprot annotation number are displayed in the Supplementary Fig. 6c) were purified by size exclusion chromatography on TSK-HW 50 (Toyopearl, 48 × 2.3 cm, flow rate of 0.5 mL/min) equilibrated with 1% ammonium acetate. Remaining maltose-1-phosphate was dephosphorylated with 10 U of alkaline phosphatase (Sigma-Aldrich) overnight at 30 °C and samples were dialyzed using Membra-Cel MC30 dialysis membrane against ultrapure water. The chain length distribution of samples was then analyzed following protocol described just above, with slight differences. Prior to APTS labeling, samples were debranched overnight at 42 °C in 50 mM sodium acetate pH 4.8 by 2 U of isoamylase from *Pseudomonas* sp. (Megazyme) and 3.5 U of pullulanase M1 from *Klebsiella planticola* (Megazyme), then desalted with AG® 501X8(D) Mixed Bed Resin. Labeled samples were diluted 10 times in ultrapure water before injection.

**Zymogram analysis**. Crude protein extracts (2 µg) or purified recombinant enzymes were loaded onto 7.5% acrylamide-bisacrylamide native gels containing 0.3% glycogen from bovine liver (Sigma-Aldrich) or 0.3% potato starch (w/v). Electrophoresis was performed in ice-cold running buffer (25 mM Tris, 192 mM glycine, 1 mM DTT) during 2 h at 120 V (15 mA per gel) using MiniProtean II (Biorad) electrophoresis system. Native gels were then soaked at room temperature and under agitation in 10 mL of incubation buffer (25 mM Tris/acetate pH 7.5, 0.5 mM DTT) supplemented with 1 mM maltose-1-phosphate or 20 mM ortho-phosphate. After overnight incubation, native gels were rinsed three times with ultrapure water prior staining with iodine solution (1% KI, 0.1% I$_2$).

**Determination of the apparent molecular weight of GlgE and TreS-Mak**. The apparent molecular weight of recombinant GlgE_EL and TreS-Mak_EL were determined using native-PAGE and gel filtration. For native-PAGE, 5, 7.5, 10, and 15% acrylamide:bisacrylamide (37.5:1) gels (20 cm × 18.5 cm ×1 mm) were loaded with 6 µg of protein of interest and some standard proteins of known mass: 15 µg carbonic anhydrase (29 kDa), 20 µg ovalbumin (43/86 kDa), 15 µg BSA (66.5/133/266/532 kDa), 15 µg conalbumin (75 kDa), 1.5 µg ferritin (440 kDa), and 25 µg thyroglobuline (669 kDa). Log10 of migration coefficient was plotted against the acrylamide concentration in the gel. Negative slopes were then plotted against molecular weights of standard proteins and the apparent molecular weight of proteins of interest was determined using slope equation. Gel permeation chromatography, superose™ 6 10/300 GL resin (30 cm × 1 cm; GE Healthcare) was equilibrated in PBS buffer (10 mM orthophosphate, 140 mM NaCl, pH 7.4) at 4 °C and with a flow rate of 0.3 mL.min$^{-1}$. Void volume was determined using Blue Dextran 2000. Standard proteins were ribonuclease A (13.7 kDa, 3 mg.mL$^{-1}$), ovalbumin (43 kDa, 4 mg.mL$^{-1}$), aldolase (158 kDa, 4 mg.mL$^{-1}$), ferritin (440 kDa, 0.3 mg.mL$^{-1}$), and thyroglobuline (669 kDa, 5 mg.mL$^{-1}$). All standard proteins were from GE's Gel Filtration Low Molecular Weight Kit and GE'S Gel Filtration High Molecular Weight Kit (GE Healthcare), except for the BSA (Sigma-Aldrich).

**Statistics and reproducibility**. The kinetic experiments and activity assays were independently replicated at least three times and individual data points were reported for each experiment. Mean, standard deviation, and fitting curves were estimated using GraphPad version 6.

**Reporting summary**. Further information on research design is available in the Nature Research Reporting Summary linked to this article.

## Data availability

All the data are available from the corresponding author on request. The chlamydial genomes used in this study including those in Supplementary Table 1 can be accessed by querying the ChlamDB database (https://chlamdb.ch/) via genome accession numbers. The Supplementary Data 1 and data sets of phylogenetic trees generated during this current study are available in the Dryad Digital Repository (https://doi.org/10.5061/dryad.8sf7m0cm7). The source data for the Figs. 5 and 8 have been included in Supplementary Data 2.

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

## Acknowledgements

The authors are very grateful to Dr Nicolas Szydlowski for providing access to the capillary electrophoresis. We also thank Dr. Agathe Subtil from Pasteur Institute (Paris) for providing *Shigella* strains and antibodies as well as the Plateforme d'Analyse des Glycoconjugues (PAGes, http://plateforme-pages.univ-lille1.fr/) for providing access to the instrumental facilities for carbohydrate analysis. This work was supported by the CNRS, the Université de Lille CNRS, and the ANR grants "Expendo" (ANR-14-CE11-0024) and "MathTest" (ANR-18-CE13-0027).

## Author contributions

M.C., S.B. and C.C. designed the project. S.B., G.G., T.P. and C.C. wrote the manuscript. T.P. carried out the genome analysis of chlamydiales. M.D. and C.K.-B. prepared the samples for electron microscopy observations. U.C., L.C. and H.T. performed phylogeny analyses. E.M., B.C. and F.B. performed proton-NMR and mass spectrometry analyses, respectively. M.C., D.K., M.D., B.H. and C.T. carried out biochemical characterizations.

## Competing interests

The authors declare no competing interests.
