## [Peer Review File · Communications Biology]

Reviewers' comments:

Reviewer #1 (Remarks to the Author):

Reviewer's Comments

Summary

In recent years, glycogen is gaining more and more attentions in a variety of research fields from archaeal lifestyles to animal aging to cancer migration, etc. In bacteria, glycogen metabolism has been extensively studied due to its close association with bacterial environmental survival and clinical pathogenicity. In addition, the study of glycogen in prokaryotes might also very likely answer the question: the origin of glycogen particles, which could contribute significantly to our understanding of the evolution of energy metabolism pathways. Previously, it was shown that all Chlamydial families maintained glycogen metabolism via the classical pathway (glgC, glgA, and glgB) except for two families Criblamydiaceae and Waddliaceae. This original study by Colpaert et al. confirmed the presence of glycogen particles in the two families via TEM, and then identified a trehalose-dependent glycogen synthesis pathway (treS-mak-glgE-glgB) in *W. chondrophila* (WC) and *E. lausannensis* (EL), which suggested that glycogen metabolism pathway is retained in all Chlamydiales without exception, hence essential to the survival and virulence of Chlamydiales.

General comments

This study is generally well designed with sufficient data to validate their conclusions. A variety of techniques were used such as bioinformatics, enzymology, molecular microbiology, and chemical analysis, etc. A systematic metabolism network for glycogen that was previously unknown in Waddliaceae and Criblamydiaceae families was finally proposed, which facilitated a better understanding of glycogen functions and metabolism in Chlamydiales. Thus, the reviewer recommend it for publication with appropriate revisions. What the reviewer also finds very interesting is that the length of glucan primers could determine the synthesis of very long chains ($DP < 4$) or generate distributive mode of chain lengths ($DP \geq 4$). How about glycogen synthase? Could it be the same scenario?

Major comments

1. Why would the authors want to include metagenomic data into the pathway analysis? Is it because there are not enough sequenced Chlamydial genomes for the analysis or because most analyzed Chlamydiales are not culturable? Due to the low quality of genomes assembled from metagenomes, phylogenetic studies of glycogen pathway distributions should be carefully interpreted.
2. Could the authors give some discussions and probably some postulations about why there are two different glycogen metabolism pathways in Chlamydiae phylum? Is there any common feature (lifestyle, host, life cycle, or niches, etc) about the five chlamydia species harbouring complete glgE-dependent glycogen pathway?
3. The information present in this manuscript is abundant and there are 10 figures in the manuscript. This is a good study. However, too much information sometimes might make the readers less interested and compromise a good story. Thus, the authors might want to consider shorten the manuscript a little bit.

Minor comments

1. In Figure 2, GlgE is classified into Class I and II. Could the author provide more information about the two groups. For example, please provide more info about the differences between the two groups, such as average length, protein domain duplication, etc? Without phylogenetic analysis, how could one distinguish between the two groups of GlgEs?
2. Page 22, line 14: what does "4TreS-4 Mak" mean?
3. Page 17: It is better to add Wc and El at the top of Figure 6 so it is easier for readers to compare the two columns of FACE results. In addition, on the right of the figure, it would be good to add corresponding annotations such as maltotriose, maltotetraose, etc.
4. Page 33, line 24: Comma is missing. Proteins were re-aligned with MUSCLE [32], block selection was carried out using BMGE...Please go through the manuscript and revise typos and small grammatical mistakes so in order to improve the language quality of the manuscript.

5. GlgP will directly generate G-1-P by degrading glycogen from the non-reducing ends. Should not this reaction be illustrated in Figure 10? By the way, are MalP and MalZ involved in the glycogen metabolism network in Waddliaceae and Criblamydiaeae families?

6. Some of the references are pretty old and there are some new studies about issues such as GlgB N-terminus, structural model of glycogen, and primers for glycogen synthesis initiation. It might be better for the authors to consider using more recent references.

Reviewer #2 (Remarks to the Author):

In this study, the authors examined glycogen biosynthesis in *Waddlia chondrophila* and *Estrella lausannensis*, two bacteria of the order Chlamydiales that have previously been believed to be unable to produce glycogen as an intracellular carbon storage compound.

The authors show that essential components of the classical glycogen pathway GlgC-GlgA-GlgB are either lacking or inactive in the analyzed strains. Enzymatic characterization of the GlgA-GlgB fusion proteins from *W. chondrophila* and *E. lausannensis* recombinantly produced in *E. coli* revealed activity of the glycogen synthase domain. However, no branching activity of the GlgB domain was observable. Additionally, bioinformatics inspection of the genomes showed a lack of GlgC necessary to provide the ADP-glucose substrate for GlgA. Altogether, this is plausible evidence that despite GlgA activity both analyzed strains cannot produce glycogen using the classical pathway.

In contrast, the authors demonstrated that the alternative GlgE pathway is functional and enables glycogen production from trehalose in the analyzed strains. For this, GlgE and the TreS-Mak fusion protein were recombinantly produced in *E. coli* and enzymatically characterized in great detail. Finally, employing EM studies of cells stained with Lugol solution, the authors demonstrate the presence of glycogen inclusions in elementary bodies of the analyzed organisms.

All experiments have been very carefully designed and conducted. The results are presented in a clear and concise way and largely support the conclusions (exception see below). However, the novelty of the presented data is rather limited. Given the substantial body of work on the GlgE pathway in actinobacteria, the present study, although dealing with a phylogenetically distant group of organisms, is largely of confirmatory nature and the results are not surprising. Somewhat novel aspects are the fact that the studied GlgE proteins have only 43% identity to the reported counterparts from *Mycobacteria* and *Streptomyces*, so it was not absolutely sure that the studied GlgE proteins indeed exhibit maltose-1-phosphate transferase activity. Furthermore, the authors demonstrated for the first time activity of a TreS-Mak fusion protein, while in actinobacteria TreS and Mak are expressed as two distinct proteins. However, in my opinion these new findings are not substantial enough to be of sufficiently broad interest to the wider audience of Communications Biology.

Finally, the authors demonstrated presence of glycogen inclusions in the cytoplasm of elementary bodies of the analyzed organisms, and the presented data strongly indicate that the observed glycogen inclusions are probably synthesized via the GlgE pathway. The authors then claim this underlines a pivotal function of storage polysaccharides in the life cycle of Chlamydiae that ensure the survival and virulence of extracellular forms. However, this is only speculative, as the current study does not provide any conclusive evidence on the role glycogen accumulation for viability or virulence of the two analyzed Chlamydiales strains. Only gene deletion mutants lacking essential elements of the GlgE pathway would be able to provide conclusive insights. Without being able to demonstrate a clear role of glycogen accumulation for viability or virulence, I believe the present study does not reach the novelty and importance to deserve publication in Communications Biology.

Reviewer #3 (Remarks to the Author):

Colpaert and co-workers through the use of genome analysis uncovered the surprising prevalence of the GlgC glycogen pathway in Chlamydiae. It has been believed that genome reduction has occurred in most intra-cellular bacterial pathogens and therefore the loss of this synthetic pathway. Furthermore, where this classical pathway has been disrupted and thought to be dysfunctional, the identification of genes that encode the recently discovered GlgE pathway has been made.

Focussing on the *Estrella lausannensis* and *Waddlia chondrophila*, the authors proceed to clone and express the suspected dysfunctional fusion genes from the GlgC and the putative genes identified from the GlgE pathways. This work uses classical biochemical techniques and biophysics, to methodically characterise and correctly annotate the function of the newly discovered GlgE genes and those fusion genes from the GlgC pathway.

This manuscript will be of interest to the pathogenic bacterium community, and importantly, increase the awareness and acceptance of the alternative GlgE glucan pathway and its role in pathogenic bacteria.

Although I am supportive of the manuscript, there are a number of minor points I would like addressing.

1) It would be useful to have the amino acid sequences of the protein expressed and purified, with tags and other additional residues, in the supporting information and UniProt accession codes.

2) There appears to be two Figure S1 and two Figure S2 in the supporting information.

3) Can the authors comment on the origin and significance of the cooperative behaviour observed with GlgE-EL during the M1P titration?

4) Although I agree with authors interpretation of the GlgE-EL elongation reaction using different length primers, it is misleading to use the term “processive” for this class of enzyme. GlgE operates via a double displacement mechanism (Ping-Pong) and therefore must release the substrate before the next catalytic cycle. Donor and acceptor molecules cannot both bind simultaneously to GlgE and form a ternary complex. Processive implies the GlgE remains bound to the growing linear chain.

5) More details in the material and methods for the kinetic assays. The concentration of enzyme used, concentrations of each substrate tested and time points if a stopped assay (i.e. malachite assay). Also the equations and fitting software used.

6) Although I understand the use of MnCl₂ gave the greatest enzyme activity for the Mak kinase, would it not have been better to use the more physiologically relevant MgCl₂ for these experiments. This does not detract from the finding that Mak is a maltose kinase, but an explanation would be good.

7) Can the authors add the source of the ¹⁴C labelled reagents used.

8) Have the authors considered that MalQ through its disproportionating activity may also be a source of maltose that can feed into the GlgE pathway?

Point-by-point response to reviewers

We thank the reviewers for their comments and their encouragement. We have revised the manuscript based on their comments and recommendations. Descriptions of specific passages we revised are provided below, and the revised text is shown in red in the manuscript.

Reviewer#1	Answers
Summary In recent years, glycogen is gaining more and more attentions in a variety of research fields from archaeal lifestyles to animal aging to cancer migration, etc. In bacteria, glycogen metabolism has been extensively studied due to its close association with bacterial environmental survival and clinical pathogenicity. In addition, the study of glycogen in prokaryotes might also very likely answer the question: the origin of glycogen particles, which could contribute significantly to our understanding of the evolution of energy metabolism pathways. Previously, it was shown that all Chlamydial families maintained glycogen metabolism via the classical pathway (glgC, glgA, and glgB) except for two families Criblamydiaceae and Waddliaceae. This original study by Colpaert et al. confirmed the presence of glycogen particles in the two families via	We thank the reviewer for positive comments. We agree also that the maintenance of glycogen metabolism pathway emphasizes an unexpected function of storage polysaccharide amongst chlamydiales.

TEM, and then identified a trehalose-dependent glycogen synthesis pathway (treS-mak-glgE-glgB) in W. chondrophila (WC) and E. lausannensis (EL), which suggested that glycogen metabolism pathway is retained in all Chlamydiales without exception, hence essential to the survival and virulence of Chlamydiales.	
General comments This study is generally well designed with sufficient data to validate their conclusions. A variety of techniques were used such as bioinformatics, enzymology, molecular microbiology, and chemical analysis, etc. A systematic metabolism network for glycogen that was previously unknown in Waddliaceae and Criblamydiaceae families was finally proposed, which facilitated a better understanding of glycogen function s and metabolism in Chlamydiales. Thus, the reviewer recommend it for publication with appropriate revisions. What the reviewer also finds very interesting is that the length of glucan primers could determine the synthesis of very long chains (DP<4) or generate distributive mode of chain lengths (DP>=4). How about glycogen synthase? Could it be the same scenario?	Although it has not been reported that bacterial glycogen synthases may switch in vitro between processive and distributive modes depending on the degree of polymerization of glucan primer, Ugalde et al., have shown that glycogen synthase of Agrobacterium tumefaciens is capable of synthesizing de novo glucans without any glucan primer. It seems reasonable to assume that the initiation of glycogen particle within bacteria cells may depends on the ability of bacterial glycogen synthase to act first as a processive enzyme for synthesizing the first long glucan (DP>20) and then switch to the distributive mode during the growth of glycogen particles.
Major comments 1. Why would the authors want to include metagenomic data into the pathway analysis? Is it because there are no enough sequenced Chlamydial genomes for the analysis or because most analyzed Chlamydiales are not culturable? Due to the low quality of genomes assembled from metagenomes, phylogenetic studies of glycogen pathway distributions should be carefully interpreted.	We had to use metagenomic-derived genomes of some members of the Chlamydiales since several families and genus level-lineages have not yet been obtained in culture and sequences only exist in such metagenomic data. Nevertheless, before using a genome derived from metagenomic data we carefully re-annotated the genomes and checked the quality by using various criteria such as the proportion of core genes, as reported previously³. For clarity, we have amended the text accordingly: Page 6 Line 13: “ It should be stressed out that several families and genus level-lineages encompass exclusively uncultured Chlamydiae species. As a consequence, derived genomes from metagenomic data have been carefully re-annotated and subjected to various quality criteria, such as the proportion of core genes as previously reported³. » 3 Pillonel T, Bertelli C, Greub G Front Microbiol 9: 1–17(2018)
2. Could the authors give some discussions and probably some postulations about why there are two different glycogen metabolism pathways in Chlamydiae phylum? Is there any common feature (lifestyle, host, life cycle, or niches, etc) about the five chlamydia species harbouring complete glgE-dependent glycogen pathway?	The GlgE pathway is present only in a few phylogenetically related chlamydial species. A reasonable assumption is that the common ancestor of the Criblamydiaceae, Waddliaceae and Parachlamydiaceae families has inherited the GlgE pathway by horizontal gene transfer. In the case of the Waddliaceae and Criblamydiaceae families, the splitting of these families was associated with genome rearrangement, which had led to a defective GlgC pathway and subsequently, to the maintenance of GlgE pathway. In the case of Parachlamydiaceae family, most of the members have lost the GlgE pathway, except for P. naegleriophila species, which have retained both pathways. Interestingly, those chlamydiales have been isolated from the aquatic amoeba Naegleria sp., which substitutes glycogen by trehalose

	biosynthesis as the main form of carbon storage. It is therefore tempting to speculate that the maintenance of GlgE is likely to offer a selective advantage by directly exploiting the trehalose of their host.
3. The information present in this manuscript is abundant and there are 10 figures in the manuscript. This is a good study. However, too much information sometimes might make the readers less interested and compromise a good story. Thus, the authors might want to consider shorten the manuscript a little bit.	We understand the reviewer's consideration. We have tried our best to limit the number of figures that cover various aspects of this study. In order to shorten the manuscript, we propose the following modifications: Page 3 Line 6: Glycogen metabolism loss appears to be a universal feature of the reductive genome evolution experienced by most if not all obligate intracellular bacterial pathogens or symbionts including Anaplasma spp., Ehrlichia spp., Wolbachia spp., Rickettsia spp. (alpha-proteobacteria), Buchnera sp., Coxiella sp. (gamma-proteobacteria), or Mycobacterium leprae (Torrabacteria). Page 3 line 14: "Glycogen synthase (GlgA) belongs to the Glycosyl-Transferase 5 family (GT5-CAZy classification) which polymerizes nucleotide-sugar into linear α-1,4 glucan. » Page 14 line 3 : To further characterize this material, time course analysis of phosphatase alkaline (PAL) treatment was performed on the reaction product suspected to be M1P obtained from sample E1 of GlgE-EL. After 180 min of incubation, the initial product is completely converted into a compound with a similar mobility than maltose (DP2) (Figure 4B). Page 22 line 5: At variance to previous GlgE expression experiments, first transformation experiment in Rosetta™ (DE3) E.coli strain did not yield colonies in spite of the absence of inducer (IPTG) (Supplementary Fig.7A). We presumed that a basal transcription of treS-mak gene associated with a substantial intracellular amount of trehalose (estimated at 8.5 mM in E.coli cell spread on Luria-Broth agar medium (Hayner et al., 2017)) lead to the synthesis of highly toxic maltose-1-phosphate. This encouraging result prompted us to perform expression of TreS-Mak protein in BL21-AI strain. This text has been removed and transferred in the methods section. Page 34 line 13 Page 24 line 15: Our preliminary investigations indicated that imidazole stabilizes or is required for the maltokinase activity domain (Supplementary Fig.7D). Hence all incubation experiments have been conducted in the presence of 125 mM of imidazole. This text has been removed and transferred in the methods section. Page 37 line 12 Page 26 line 20: This chloroplastic UDP-glucose pyrophosphorylase activity, dedicated to sulfolipid biosynthesis belongs to a set of 50 to 90 chlamydial genes identified, as lateral gene transfer, in the genomes of Archaeplastida.

Minor comments	
1. In Figure 2, GlgE is classified into Class I and II. Could the author provide more information about the two groups. For example, please provide more info about the differences between the two groups, such as average length, protein domain duplication, etc? Without phylogenetic analysis, how could one distinguish between the two groups of GlgEs?	Sequence alignments did not emphasize any particular gaps, specific sequences or specific lengths between classes. These specific features are likely to arise when GlgE sequences are compared to other carbohydrate enzyme activities belonging to GH13 superfamily (e.g. branching enzymes, α-amylase). The distinction between classes I and II requires a more detailed analysis.
2. Page 22, line 14: what does "4TreS-4 Mak" mean?	In mycobacteria, TreS and Mak proteins form an octo-heterotetramic complex composed of 4 subunits of TreS and 4 subunits of Mak. We clarified the meaning of 4TreS-4 Mak in the manuscript.
3. Page 17: It is better to add Wc and EI at the top of Figure 6 so it is easier for readers to compare the two columns of FACE results. In addition, on the right of the figure, it would be good to add corresponding annotations such as maltotriose, maltotetraose, etc	For sake of clarity, we have included the annotation for each FACE analysis as suggested by the reviewer.

4. Page 33, line 24: Comma is missing. Proteins were re-aligned with MUSCLE [32], block selection was carried out using BMGE... Please go through the manuscript and revise typos and small grammatical mistakes so in order to improve the language quality of the manuscript.

The manuscript has been revisited for grammatical mistakes.

5. GlgP will directly generate G-1-P by degrading glycogen from the non-reducing ends. Should not this reaction be illustrated in Figure 10? By the way, are

The figure 10 has been updated to highlight the GlgP activity. Chlamydial genomes have been probed using either GlgP or malP sequences of

MalP and MalZ involved in the glycogen metabolism network in Waddliaceae and Criblamydiaceae families?	E.coli, the blast searches reveal that most chlamydiales encompass a single glycogen phosphorylase with some exception as shown in figure 1. Furthermore, blast search did not evidence any homologous sequence to MalZ.
6. Some of the references are pretty old and there are some new studies about issues such as GlgB N-terminus, structural model of glycogen, and primers for glycogen synthesis initiation. It might be better for the authors to consider using more recent references.	Two recent references have been added to the manuscript: The work of Wang et al., on the role of N-terminus domain in GlgB (1) and a recent review about glycogen metabolism in bacteria (2). However, based on this review as well as on the basis of our search on PubMed database, no recent publications have been cited on either the structural characterization of glycogen or the glycogen initiation other than those indicated in the manuscript. (1) Wang et al., BMC Microbiol. 15, 96 (2015) (2) Cifuentes et al., Biochem J. 476, 2059-2092 (2019)
Reviewer#2	Answers
All experiments have been very carefully designed and conducted. The results are presented in a clear and concise way and largely support the conclusions (exception see below). However, the novelty of the presented data is rather limited. Given the substantial body of work on the GlgE pathway in actinobacteria, the present study, although dealing with a phylogenetically distant group of organisms, is largely of confirmatory nature and the results are not surprising. Somewhat novel aspects are the fact that the studied GlgE proteins have only 43% identity to the reported counterparts from Mycobacteria and Streptomyces, so it was not absolutely sure that the studied GlgE proteins indeed exhibit maltose-1-phosphate transferase activity. Finally, the authors demonstrated presence of glycogen inclusions in the cytoplasm of elementary bodies of the analyzed organisms, and the presented data strongly indicate that the observed glycogen inclusions are probably synthesized via the GlgE pathway. The authors then claim this underlines a pivotal function of storage polysaccharides in the life cycle of Chlamydiae that ensure the survival and virulence of extracellular forms. However, this is only speculative, as the current study does not provide any conclusive evidence on the role glycogen accumulation for viability or virulence of the two analyzed Chlamydiales strains. Only gene deletion mutants lacking essential elements of the GlgE pathway would be able to provide conclusive insights. Without being able to demonstrate a clear role of glycogen accumulation for viability or virulence, I believe the present study does not reach the novelty and importance to deserve publication in Communications Biology.	The originality of this work is that all obligate intracellular bacteria belonging to chlamydia phylum have retained a storage polysaccharide pathway through the bottleneck of genome reduction and rearrangement. The loss of glycogen metabolism is a common occurrence and is even considered as a marker in obligate intracellular bacteria. Of course, the functional characterization of mutants would undoubtedly highlight the mandatory role glycogen in the life cycle of chlamydia. For this reason, we have tuned down our conclusion by rephrasing the sentence in the discussion section: Page 31 line 18: “The preservation of glycogen metabolism pathway through the bottleneck of genome reduction process shed light on a pivotal suggests an unexpected function of glycogen that has been hitherto underestimated within Chlamydiae »
Reviewer#3	Answers
Although I am supportive of the manuscript, there are	To address this comment, we have amended the supplementary figures S2, S6 and S7 with both

a number of minor points I would like addressing. 1) It would be useful to have the amino acid sequences of the protein expressed and purified, with tags and other additional residues, in the supporting information and UniProt accession codes.	amino acid sequences of the recombinant proteins and Uniprot accession numbers.
2) There appears to be two Figure S1 and two Figure S2 in the supporting information.	The first two documents in the supplementary data are table S1 and S2 and then, figures S1 and S2. Supplementary data have been renamed as Supplementary Table and supplementary figure to improve clarity.
3) Can the authors comment on the origin and significance of the cooperative behaviour observed with GlgE-EL during the M1P titration?	The cooperative behavior suggests additional M1P binding sites on homo-dimers of GlgE. This allosteric behavior may indicate as M1P molecules act as positive effector of GlgE activity.
4) Although I agree with authors interpretation of the GlgE-EL elongation reaction using different length primers, it is misleading to use the term “processive” for this class of enzyme. GlgE operates via a double displacement mechanism (Ping-Pong) and therefore must release the substrate before the next catalytic cycle. Donor and acceptor molecules cannot both bind simultaneously to GlgE and form a ternary complex. Processive implies the GlgE remains bound to the growing linear chain.	We understand the concern of reviewer that processive mode is not compatible with Ping-Pong mechanism. We have replaced the term “processive” by “processive-like” and have added an explanation in the manuscript. Page 18 line 25: « The processive behavior of GlgE enzymes was unexpected since GlgE activity has been reported to operate a double displacement reaction (i.e. Ping-Pong mechanism) involving the release of (2+n) glucan prior the next reaction¹⁸ »
5) More details in the material and methods for the kinetic assays. The concentration of enzyme used, concentrations of each substrate tested and time points if a stopped assay (i.e. malachite assay). Also the equations and fitting software used.	We have included more details in the methods section as suggested.
6) Although I understand the use of MnCl2 gave the greatest enzyme activity for the Mak kinase, would it not have been better to use the more physiologically relevant MgCl2 for these experiments. This does not detract from the finding that Mak is a maltose kinase, but an explanation would be good.	We agree with the reviewer that to some extent enzymatic characterization should be studied under physiological conditions. The range of intracellular concentrations of Mn²⁺ varies from 5 to 35 μM in the free-living Escherichia coli (1), which is much lower that concentrations used in this study (1mM to 10 mM). It is very likely that we might not measure any Mak activity under our experimental conditions. (1) Martin et al., (2015) Plos Genetics DOI:10.1371/journal.pgen.100497.
7) Can the authors add the source of the 14C labelled reagents used.	We have indicated the source of 14C-nucleotide-sugars in the methods.

8) Have the authors considered that MalQ through its disproportionating activity may also be a source of maltose that can feed into the GlgE pathway?	We did not consider that MalQ activity could be a source of maltose because it is widely accepted that both MalQ and phosphorylase activities act in synergy to completely metabolize short malto-oligosaccharides (DP<4). In addition, detailed characterizations of MalQ activity from various sources have shown that maltose molecules are scarcely produced during disproportionating reactions even if maltotriose is used as substrate.
---	---

REVIEWERS' COMMENTS:

Reviewer #1 (Remarks to the Author):

I would like to thank the authors for their efforts in addressing all my comments and concerns. After going through the point-to-point response in the rebuttal letter and the revised manuscript, I believe that they have improved the manuscript by comprehensively considering the issues and satisfactorily addressing them in this revision. Thus, I strongly recommend the publication of the manuscript.

Liang Wang, PhD
Xuzhou Medical University
Xuzhou, Jiangsu Province
221000, China

Reviewer #3 (Remarks to the Author):

Colpaert and co-workers have performed a thorough biochemical and bioinformatical study of the glucan synthetic pathways present in Chlamydiae. The conclusion from the study was that where the classical GlgC pathway is dysfunctional, the alternative GlgE pathway was in operation. This is more surprising as it is believed that most intra-cellular bacteria have undergone genome reduction losing the ability to synthesise glucan.

It is important to correctly determine the function of gene products using biochemical techniques, rather than relying on gene annotations. This is exemplified with GlgE wrongly being annotated as an alpha amylase.

As important as biochemically confirming the presence of the GlgE pathway, Colpaert et al also isolated and tested elements of the GlgC pathway and showed it to be non-functional.

This study is important and contributes greatly to the field, as recognition of the prevalence and significance of the GlgE pathway in bacteria has been slow in the last decade.

It is hoped that with more manuscripts like these, the GlgE pathway will be discussed as freely as the classical pathway.

On the comment of M1P cooperativity being a result of an allosteric M1P binding site, a conformational change within the GlgE dimer could also account for the observation without the requirement of allostereism. Both explanations would require further experiments beyond the scope of this manuscript.

I feel the authors have address the reviewers comments thoroughly and I am supportive of the manuscript for publication in its current form.

Yours faithfully,

Karl Syson.